# ON ERRONEOUS AGREEMENTS OF CLIP IMAGE EMBEDDINGS

## ABSTRACT

Recent research suggests that the failure of Vision-Language Models (VLMs) in visual reasoning could be attributed to the CLIP image encoder ambiguously encoding distinct images into embeddings with high cosine similarity, namely *erroneous agreements*. In this paper, we show that they are not the sole issue, as multimodal large language models (MLLMs) may extract distinct information even from image embeddings with high cosine similarities. On Subset A of the What'sUp benchmark, where the Left/Right image pairs are embedded by CLIP with average cosine similarity greater than 0.99, CLIP's performance is near random guess. In contrast, LLaVA-1.5-7B, which uses the same image encoder as CLIP, achieves nearly 100% accuracy. This discrepancy is also observed between LLaVA-1.5-7B and CLIP-like models on similar benchmarks. To investigate this performance gap, we conduct controlled experiments to test the effect of varying evaluation methods, training data, and language processing choices. We find that the CLIP image embeddings contain more extractable information than previously suggested, but it is likely obscured by the inadequate vision-language alignment of the CLIP's paradigm. Motivated by this observation, we reconsider the LLaVA-1.5 model on the MMVP benchmark, for which prior work showed that it could not distinguish image pairs with high cosine similarity. We observe a performance gain brought about by an alternative decoding algorithm, which attends more to visual input. Further, we show that the accuracy significantly increases if the model can take both images as input to emphasize their nuanced differences. Both findings indicate that LLaVA-1.5 did not utilize extracted visual information sufficiently. In conclusion, our findings suggest that while improving image encoders could benefit VLMs, there is room to enhance the models with a fixed image encoder through better strategies for extracting and utilizing visual information.

## 1 INTRODUCTION

Despite the rapid development and success of Vision-Language Models (VLMs), recent work pointed out that state-of-the-art VLMs (Radford et al., 2021; Zhai et al., 2023; Liu et al., 2024; Google, 2023a;b; OpenAI, 2023) still struggle with some simple visual reasoning tasks (Li et al., 2023d; Liu et al., 2023b; Tong et al., 2024c; Rahmanzadehgervi et al., 2024), where they were asked to answer questions about the images, such as recognizing shapes and describing the object relationships. These are basic tasks that VLMs should be able to solve before we deploy them to real-world scenarios like home robots responding to spoken or written commands.

Recent work argued that the pretrained CLIP image encoder (Radford et al., 2021), which serves as the "eyes" of many VLMs, could be the cause and cure for such visual shortcomings (Tong et al., 2024c). In these VLMs, any input image is first encoded by the CLIP image encoder and then used to calculate image-text similarity or as the input for a generative language model. Therefore, any deficiency of the CLIP image encoder propagates into the VLMs. The deficiency found by the authors is named *erroneous agreements*: Visually different images could be ambiguously encoded with high cosine similarity in the embedding space. They claimed this suggested information loss and caused VLMs' failure in relevant visual reasoning tasks, such as the MMVP benchmark (Tong et al., 2024c). This benchmark consists of selected, semantically distinct image pairs erroneously agreeing in the CLIP image embedding space, and CLIP-based VLMs failed to answer questions

regarding the visual semantic difference better than random chance. This criterion is adopted in Taghipour et al. (2024) and similarly used in Tong et al. (2024b).

In this work, we provide evidence that VLMs face challenges beyond erroneous agreements. The query-relevant visual information might still be present in the image embeddings despite the high cosine similarity, but a better strategy is required to pull it out. For instance, in the What'sUp benchmark (Kamath et al., 2023a) with paired, tightly controlled image pairs for evaluating VLM's spatial reasoning ability, the average cosine similarity of image pairs on three out of four subsets is greater than 0.95 in CLIP image embedding space, reaching the similarity threshold in Tong et al. (2024c). While CLIP's accuracy in distinguishing these images is nearly random (about 50%), LLaVA-1.5-7B (Liu et al., 2024) with the pretrained, frozen image encoder of CLIP-ViT-L/14-336px still achieves beyond 80% in binary classification accuracies on all four subsets in What'sUp. Similarly, on the COCO-spatial and GQA-spatial benchmark used in Kamath et al. (2023a), LLaVA-1.5-7B surpasses CLIP-like models (including SigLIP (Zhai et al., 2023)) by a large margin. On the more challenging MMVP and MMVP-VLM benchmark (Tong et al., 2024c), though its absolute performance is poor, LLaVA-1.5 still outperforms CLIP-like models, showcasing a stronger ability to extract information from given image embeddings.

What causes their discrepancy in extracting the given visual information? First, we unify the evaluation methods of CLIP and LLaVA and observe that the performance gap still exists. Then we decompose their difference into three parts: training data, language processing choice, and model paradigm (training and inference pipeline). Through ablation studies, we find that CLIP's failure is likely caused by the inadequate visual-language alignment of CLIP's paradigm. This also implies that the visual information extraction module in LLaVA-1.5, consisting of the two-layer MLP connector and the language model, adopts an inherently different mechanism from CLIP's paradigm.

The above results emphasize the importance of effective visual information extraction and highlight LLaVA-1.5's extracting ability. However, its poor performance on the MMVP benchmark remains a mystery. We look into its failure and provide insight into future directions in the discussion section. To help LLaVA-1.5 keep the visual information during decoding, we try an alternative decoding algorithm, Multi-Modal Mutual-Information Decoding (M3ID) (Favero et al., 2024), leading to performance gain (+6%). We further find that visual nuances are often extracted and aligned with the correct semantics by LLaVA-1.5 rather than being discarded after visual encoding, but they did not induce enough difference in outputs. To explore the amount of such visual formation, we reevaluate LLaVA-1.5 with relaxed constraints, which allows for comparing the slight difference induced in the outputs of two images. In this setting, its accuracy is significantly above random chance (+23.3%), while the result in the original one-image setting is just around random chance (+0.3%), suggesting insufficient visual information utilization in the original evaluation. In conclusion, despite the erroneous agreements in the CLIP embedding space, visual nuances might still be extracted with improved strategies. This underscores the potential to enhance model performance by employing better extraction and utilization techniques with the same pretrained image encoder.

## 2    RELATED WORK

**Benchmarking VLMs' visual reasoning ability.** Vision reasoning tasks underline VLMs' visual perception ability. Many recent challenging benchmarks on visual reasoning focus on assessing specific abilities of current VLMs like compositionality (Winoground (Thrush et al., 2022), ARO (Yuksekgonul et al., 2023), SugarCrepe (Hsieh et al., 2024)), hallucination (POPE (Li et al., 2023d), HallusionBench (Liu et al., 2023a), and VHILT (Rawte et al., 2024)), distinguishing image pairs (MMVP (Tong et al., 2024c)), spatial understanding of VLMs (What'sUp (Kamath et al., 2023a) and Embspatial-bench (Du et al., 2024)), and core visual perception abilities (BLINK (Fu et al., 2024) for various aspects like visual correspondence and BlindTestbasic (Rahmanzadehgervi et al., 2024) for recognizing basic patterns). State-of-the-art VLMs often fail unexpectedly on simple test cases, performing significantly worse than human accuracy or even random guess. For our discussion on erroneous agreements, we mainly consider the MMVP and What'sUp benchmark since their image pairs exhibit this property in the CLIP embedding space. Nevertheless, findings on these benchmarks reveal the relationship between vision encoders and VLMs, supporting the broader goal of enhancing VLMs for general visual reasoning.

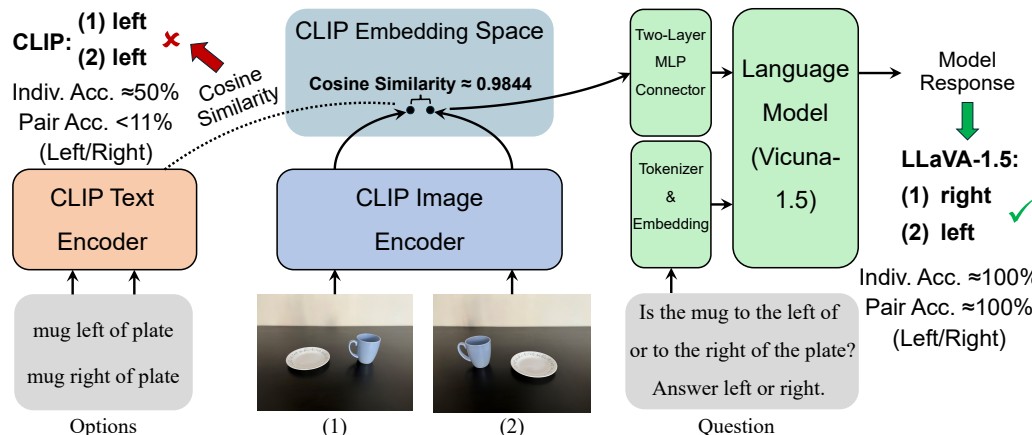

Figure 1: An illustration of CLIP and LLaVA-1.5 model structures sharing the same pretrained image encoder with an example test case from the Left/Right subset of What'sUp benchmark. We find that the query-relevant nuances in the CLIP image embeddings may be extracted by LLaVA-1.5 despite erroneous agreements, and we note their performance gap on several similar benchmarks.

**Exploring causes of the visual shortcomings of VLMs.** Researchers are actively exploring the root causes of VLMs' failures on benchmarks above, mainly from the model perspective. **(1) Vision modality.** Tong et al. (2024c) argued that the flawed CLIP image encoder adopted by many VLM architectures could lead to downstream failure because of erroneous agreements. (Chandhok et al., 2024) agreed that the image encoder is responsible for the information loss in spatial reasoning tasks since CLIP's performance is quite low. However, we find evidence that erroneous agreements do not necessarily lead to VLM's failure and that LLaVA is stronger at extracting information from similar visual embeddings than CLIP. **(2) Language understanding.** In CLIP, Kamath et al. (2023b); Tong et al. (2024b) found that its text encoder could also lose relevant information during encoding. In multimodal LLMs (MLLMs), the language model might not timely terminate answer generation (Yue et al., 2024), put false priority on the input text and format (Stan et al., 2024), or neglect information like negation (Quantmeyer et al., 2024). Qiao et al. (2024) decoupled the perception stage and reasoning stage of VLMs and found that they are often limited by reasoning ability. In this paper, we perform an ablation study on the text encoder and find that CLIP-like models still fail when equipped with a stronger text encoder. **(3) Vision-language alignment.** Others discussed the importance of modality alignment, such as visual grounding (Rajabi & Kosecka, 2023). From the language model side, Ye et al. (2024) pointed out that the MLLMs might utilize multimodal spurious correlation in the training data due to the coarse-grained training objectives. Similarly, Yang et al. (2024a) found that the model wrongly raised the probability of deceptive candidates. From the image encoder side, Yang et al. (2024b) proposed the cross-modal Alignment and Correspondence score of visual representations, which is linearly correlated to model performance. We abstractly view the components of VLMs other than the vision part as visual information extraction and utilization module and demonstrate their different abilities. **(4) Other factors.** Apart from the model, others looked into the problem with training data (Udandarao et al., 2024) or downstream tasks, such as the hardness of the visual query (Zhang et al., 2024).

**Improving the visual reasoning ability of VLMs.** Following the observations about VLM's limitation, researchers mainly focused on improving the model structure with better image encoders and vision-languageconnectors (Luo et al., 2024a; Zeng et al., 2021; Yao et al., 2024; Kar et al., 2024; Jiang et al., 2023; Zong et al., 2024; Xu et al., 2024; Tong et al., 2024a; Meng et al., 2024) or using different training objectives with additional loss terms (Zhang et al., 2023; Zeng et al., 2024). From the data-centric perspective, previous paper tried adding relevant instruction tuning data (Ranasinghe et al., 2024; Chen et al., 2024), using long caption in pretraining (Zheng et al., 2024), hard negative mining (Yuksekgonul et al., 2023; Paiss et al., 2023) or using synthesized images (Chatterjee et al., 2024; Jiao et al., 2024).

Post-training techniques are also explored to improve the performance of off-the-shelf VLMs. Some leveraged feedback from other models (Wang et al., 2024a; Luo et al., 2024b; Deng et al., 2024), various text or visual prompting methods (Wan et al., 2024; Lei et al., 2024; Wu et al., 2024), or multi-turn reflection (Huang et al., 2024; Wu et al., 2024; Wu & Xie, 2023; Kim et al., 2024b).

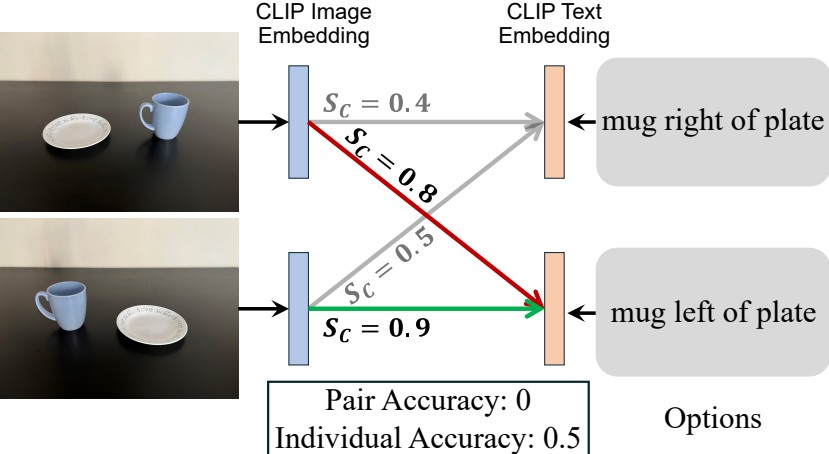

Figure 2: Example for evaluating CLIP-like models on What'sUp benchmark. For two-way evaluation, a test case consists of two similar images and two captions. The model chooses one caption for each image, and it gets one point in pair accuracy only if choosing correctly for both images.

Without external feedback or changing the task format, others developed probability-based output correction (Zhou et al., 2023), including hallucination-reducing decoding strategies (Chuang et al., 2023; Yang et al., 2024a; Kim et al., 2024a;c; Favero et al., 2024). Besides decoding, visual attention recalibration was proposed in response to the false priority put by VLMs (Woo et al., 2024). In this work, we achieve performance gain on MMVP through a decoding algorithm, M3ID (Favero et al., 2024), and test a new evaluation with relaxed constraints to show that current visual information utilization in LLaVA-1.5 is insufficient.

## 3 ERRONEOUS AGREEMENTS

We begin by introducing the task setup and the concept of erroneous agreements. Using a toy example, we demonstrate that information might still be extracted through alternative methods despite erroneous agreements. This is further validated through LLaVA-1.5's good performance on the What'sUp benchmark (Kamath et al., 2023a), showing that erroneous agreements are not the sole issue as it is possible for LLaVA-1.5 to extract distinct visual information from highly similar embeddings. We also notice a significant performance gap between LLaVA-1.5 and CLIP on What'sUp and across several other benchmarks.

### 3.1 TASK SETUP

This paper focuses on the setup in which VLMs are asked to choose from several captions based on a given image. For MLLMs, the image is accompanied by a question. Here, we use What'sUp benchmark Kamath et al. (2023a), which was proposed for evaluating VLM's spatial reasoning ability. Every test case includes four captions (e.g., "A dog left of/right of/on/under a table") and four corresponding images photographed with minimal change except for the object spatial relationship. For the convenience of calculating cosine similarity and comparing it to model performance, we split each test case into two pairs: In the previous example, one pair consists of "A dog left of a table" and "A dog right of a table" together with the ground truth images, and the other pair is the remaining captions and images. This way, we get four subsets of the original benchmark.

For CLIP-like models, we calculate the matching score between images and texts. For CLIP with image encoder $f_v$ and text encoder $f_t$, this is the cosine similarity between its image embeddings $f_v(\mathbf{v})$ and text embeddings $f_t(\mathbf{t})$, denoted as

$$S_C(f_v(\mathbf{v}), f_t(\mathbf{t})) = \frac{f_v(\mathbf{v})^\top f_t(\mathbf{t})}{||f_v(\mathbf{v})||||f_t(\mathbf{t})||} \tag{1}$$

As evaluation metrics, **pair accuracy** (Tong et al., 2024c; Kamath et al., 2023a) requires correct matching for both images, while the accuracy for two images independently is called **individual**

**accuracy**. An example from the What'sUp benchmark, together with the evaluation of CLIP, is shown in Figure 2.

The concept of erroneous agreements stems from this evaluation setup for CLIP. Specifically, if erroneous agreement happens for two different images $\mathbf{v_1}$ and $\mathbf{v_2}$, then

$$S_C(f_v(\mathbf{v_1}), f_v(\mathbf{v_2})) > \tau \tag{2}$$

where $\tau$ is a chosen threshold near 1 (e.g., $\tau = 0.95$ is used in Tong et al. (2024c)). Intuitively, when the cosine similarity is high, the two image embeddings point in nearly the same direction, and they will be close in Euclidean distance after $l_2$-normalization. From the view of captions, they will result in highly similar image-text matching scores with any caption. From the image side, the margin is small, so adding a slight noise in either embedding could reverse the preference over the captions, which might result from a small perturbation in either $\mathbf{v_1}$ or $\mathbf{v_2}$. Hence, the difference between $\mathbf{v_1}$ and $\mathbf{v_2}$ cannot be stably extracted by the CLIP model, and the relevant information seems lost. If this suggests the CLIP image encoder is "blind," this will also undermine VLMs that use it as "eyes."

This intuition is supported by the results on the MMVP benchmark designed to include image pairs with a cosine similarity greater than 0.95 for CLIP embeddings but less than 0.6 for DINOv2 embeddings, along with the MMVP-VLM benchmark for CLIP-like models (Tong et al., 2024c). LLaVA-1.5 and CLIP perform close to random chance on these two benchmarks, respectively. The authors also demonstrated a correlation between CLIP model accuracy and LLaVA-1.5's accuracy on different visual patterns.

### 3.2 DO ERRONEOUS AGREEMENTS MEAN BLINDNESS?

We note that cosine similarity does not depict all aspects of vector pairs. One criticism of it as the similarity metric is that it only captures the linear relationship of vectors. As an example, consider the following image embeddings

$$f_v(\mathbf{v_1}) = [10, 11, 12]^\top, f_v(\mathbf{v_2}) = [12, 11, 10]^\top$$

While $S_C(f_v(\mathbf{v_1}), f_v(\mathbf{v_2})) > 0.989$, Spearman's rank correlation coefficient can tell their sharp difference: $\rho = -1$, showing that their order information is fully opposed. Therefore, the difference in visual inputs might still be extracted through other means when erroneous agreements occur.

We show that this scenario happens in experiments. In many VLMs using CLIP image encoder as their "eyes," the output score is nonlinear, different from CLIP-like models. For instance, in LLaVA-1.5 (Liu et al., 2024), the CLIP image embeddings first pass through a two-layer MLP and are then used as input tokens for the transformer, Vicuna-1.5, which yields the token probability determining the model response. We evaluate LLaVA-1.5-7B using the pretrained weights and design the question format. (An illustration and an example are in Figure 1, and more details are in Appendix A.1.) The results are shown in Table 1. Despite the high cosine similarity, LLaVA-1.5-7B's individual accuracy and pair accuracy are both quite high, showing that it can extract and align query-relevant information from image embeddings and produce the correct answer. In other words, erroneous agreements do not contribute to the failure of VLMs on their own.

Apart from this two-way evaluation, in Table 2, we report the results of the original evaluation, which is a four-way classification for each image. Besides, we include the results on COCO-spatial and GQA-spatial used in Kamath et al. (2023a) also for evaluating VLM's spatial reasoning ability. These benchmarks are in the format of an image paired with two captions differing only by a preposition. On all these benchmarks, LLaVA-1.5-7B wins CLIP-like models by a large margin, even compared with the best model XVLM-COCO (Zeng et al., 2021) reported in the paper. We also find that this performance gap relative to CLIP generalizes to some other MLLMs with different scales and language models in Appendix B.5.

To see if LLaVA-1.5-7B shows better extraction ability on tasks other than recognizing spatial relationships, we compare it and CLIP on MMVP and MMVP-VLM (Tong et al., 2024c). There was no direct comparison in the original paper: The MMVP benchmark is not in CLIP's format, while the MMVP-VLM benchmark is incompatible with MLLMs. So, we manually convert them into suitable formats without changing the content, and the evaluation of CLIP on MMVP-VLM is changed to the method described in Section 3.1 accordingly. The results are shown in Table 3. Although their absolute accuracy is low, there is a clear performance gap between CLIP-ViT-L/14-336px and LLaVA-1.5-7B with the same image encoder.

Table 1: The average cosine similarity of CLIP-ViT-L/14-336px embeddings and results of LLaVA-1.5-7B model on four subsets of What'sUp. The individual accuracy and pair accuracy are in percentage points. The average cosine similarity of the CLIP-ViT-L/14-336px image embeddings for image pairs is calculated for each category.

| | What'sUp Subset A | | | | What'sUp Subset B | | | |
| | Left/Right | | On/Under | | Left/Right | | Front/Behind | |
| | Indiv. | Pairs | Indiv. | Pairs | Indiv. | Pairs | Indiv. | Pairs |
|---|---|---|---|---|---|---|---|---|
| CLIP-ViT-L/14-336px | 49.0 | 1.9 | 61.7 | 23.3 | 54.9 | 10.8 | 51.5 | 7.8 |
| LLaVA-1.5-7B | **99.0** | **98.1** | **80.1** | **60.2** | **100** | **100** | **98.5** | **97.1** |
| Avg. Embedding Cosine Sim. | 0.995 | | 0.971 | | 0.955 | | 0.902 | |

Table 2: Results of varied vision-language models on What'sUp, COCO-spatial, and GQA-spatial benchmark. We test the models on the four-way classification of each image. "Set of 4" is the correctness for all four images in a set.

| | What'sUp Subset A | | | What'sUp Subset B | | | COCO-spatial | | GQA-spatial | |
| | Indiv. | Pairs | Set of 4 | Indiv. | Pairs | Set of 4 | One-obj. | Two-obj. | One-obj. | Two-obj. |
|---|---|---|---|---|---|---|---|---|---|---|
| CLIP-ViT-L/14-224px | 26.7 | 1.0 | 0.0 | 25.7 | 1.5 | 0.0 | 49.1 | 50.2 | 46.0 | 48.1 |
| CLIP-ViT-L/14-336px | 28.9 | 1.0 | 0.0 | 27.2 | 1.0 | 0.0 | 48.9 | 51.1 | 46.6 | 49.1 |
| SigLIP-ViT-L/16-384px | 26.7 | 0.0 | 0.0 | 28.7 | 2.0 | 0.0 | 50.3 | 48.6 | 47.8 | 48.7 |
| XVLM-COCO | 41.8 | 17.0 | 1.9 | 42.2 | 15.7 | 2.9 | 68.4 | 73.6 | 69.1 | 67.0 |
| LLaVA-1.5-7B | **62.1** | **41.3** | **14.6** | **74.0** | **61.8** | **23.5** | **96.0** | **82.3** | **96.0** | **90.7** |
| Random chance | 25.0 | 6.3 | 0.4 | 25.0 | 6.3 | 0.4 | 50.0 | 50.0 | 50.0 | 50.0 |

## 4 INVESTIGATE THE PERFORMANCE GAP

The above performance gap might be caused by various factors: evaluation methods, training data, language processing choice, and model paradigm. Firstly, we used cosine-similarity-based evaluation for CLIP-like models and model-response-based evaluation for LLaVA-1.5. Apart from this, CLIP-like models and LLaVA-1.5 were trained on different data, adopted different language processing techniques, and were in different VLM paradigms.

In this section, we design and conduct ablation studies to determine whether these factors contribute to the failure of CLIP-like models. The ablation of evaluation methods is conducted first to determine whether there is really a performance gap. Then, we control the training data and text encoder of CLIP-like models to see if they cause the failure (See the illustration in Figure 3).

### 4.1 UNIFIED EVALUATION

One might first question the different evaluations performed on the CLIP-like models and LLaVA-1.5. For the former, the evaluation is numeric-value based, while the latter is judged by its output response as a conversation agent, following the practice in previous work (Tong et al., 2024c). Hence, we test LLaVA-1.5 again using standard Multiple-Choice (MC) evaluation, where we rank the perplexity of options ("A" and "B") calculated based on the probability of output tokens. This is similar to the CLIP-like models' evaluation, where the image-text matching scores are ranked.

In Table 3, we observe that MC evaluation yields similar results on the MMVP benchmark and even better results on MMVP-VLM. Thus, this does not count for their performance discrepancy. The increase in results on MMVP-VLM is possibly related to the hallucination of MLLMs, where they tend to follow their language prior during long answer generation and might gradually "forget" the visual input.

### 4.2 TRAINING DATA

The web-crawled image-caption corpora used for pretraining models like CLIP generally contain very few high-quality, unambiguous image-caption pairs with prepositions, as pointed out in Kamath

Table 3: Results of CLIP and LLaVA-1.5 on the MMVP and MMVP-VLM benchmark. The last row is the LLaVA-1.5-7B accuracy in the Multiple-Choice setting. Accuracy with an asterisk is obtained on a converted version of the original benchmark.

|  | MMVP | | MMVP-VLM | |
|  | Indiv. | Pairs | Indiv. | Pairs |
| --- | --- | --- | --- | --- |
| CLIP-ViT-L/14-336px | 54.7* | 14.0* | 58.1 | 20.7 |
| LLaVA-1.5-7B | **61.7** | **25.3** | 60.7* | 28.2* |
| LLaVA-1.5-7B(MC) | 61.3 | **25.3** | **63.7*** | **31.1*** |
| Random chance | 50.0 | 25.0 | 50.0 | 25.0 |

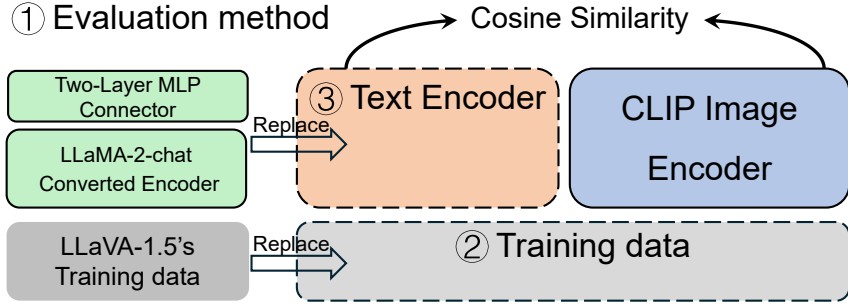

Figure 3: Illustration for the CLIP paradigm and the ablation studies.

et al. (2023a). Equipped with the pretrained CLIP image encoder, LLaVA-1.5 was finetuned on LCS-558K, a subset from LAION-CC-SBU with BLIP captions, and Instruction-following Data Mixture (hereafter referred to as DataMix-665K) (Liu et al., 2024). These datasets were carefully curated and thus of higher quality than web-crawled data. Besides, they are more relevant to the spatial reasoning tasks: They have around 13K samples with phrases containing "left" or "right," accounting for around 1% of all data. This ratio is higher than that in LAION-2B (English), where various prepositions, including other directions like top and bottom, represent less than 0.22% of the training data (Kamath et al., 2023a). Hence, we hypothesized that LLaVA-1.5's visual information extraction ability benefits from these data.

To check the effect of training data, we use LLaVA-1.5's training data to fine-tune CLIP-like models. We convert both datasets to the image-caption format (More details in Appendix B.2). By default, we lock the image encoder during finetuning for strict ablation. The results are shown in Table 4. Finetuning on LLaVA-1.5's training data slightly improves CLIP's performance, but it does not help SigLIP. Still, their accuracy is around random chance. On SigLIP, we try unlocking the image encoder during finetuning, but this does not increase model performance notably either (See results in Appendix B.3). In Appendix B.4, we also explore the effect of high-quality data on the gap between LLaVA-1.5 and another VLM paradigm, XVLM (Zeng et al., 2021), and find that data do not explain it solely. This result aligns with the failure in previous work to enhance CLIP models significantly by finetuning them on a much larger, preposition-focused subset of LAION (Kamath et al., 2023a).

One might argue that the CLIP training objective differs from LLaVA-1.5, heavily relying on negative samples beyond data quality, so CLIP's failure on the new dataset might be due to the lack of corresponding negatives. Next, we check this by observing whether negative samples help the CLIP-like models learn better. For this experiment, we focus on the model's ability to distinguish "left" and "right" and use the Left/Right subsets as the benchmarks. We construct hard negative captions by switching the related phrases to their opposite, e.g., replacing "on the left" with "on the right." The loss objective changes accordingly, following the NegCLIP method (Yuksekgonul et al., 2023).

The results are shown in Table 5. This strategy does not increase the model performance consistently on the Left/Right subset, which is observed in Kamath et al. (2023a) as well. Likewise, we try unlocking the image encoder of SigLIP in this setting, which does not make a big difference (See

Table 4: Results of CLIP and SigLIP on What'sUp, COCO-spatial, and GQA-spatial benchmark after finetuning on LLaVA-1.5's training data.

| | What'sUp Subset A | | | What'sUp Subset B | | | COCO-spatial | | GQA-spatial | |
| | Indiv. | Pairs | Set of 4 | Indiv. | Pairs | Set of 4 | One-obj. | Two-obj. | One-obj. | Two-obj. |
|---|---|---|---|---|---|---|---|---|---|---|
| CLIP-ViT-L/14-336px | 28.9 | 1.0 | 0.0 | 27.2 | 1.0 | 0.0 | 48.9 | 51.1 | 46.6 | 49.1 |
| + finetuning | 31.1 | 9.7 | 0.0 | 30.6 | 7.4 | 0.0 | 54.2 | 55.5 | 51.0 | 52.6 |
| SigLIP-ViT-L/16-384px | 26.7 | 0.0 | 0.0 | 28.7 | 2.0 | 0.0 | 50.3 | 48.6 | 47.8 | 48.7 |
| + finetuning | 27.2 | 1.9 | 0.0 | 24.8 | 2.0 | 0.0 | 49.3 | 50.5 | 47.0 | 54.6 |
| Random chance | 25.0 | 6.3 | 0.4 | 25.0 | 6.3 | 0.4 | 50.0 | 50.0 | 50.0 | 50.0 |

Table 5: Two-way evaluation results of CLIP and SigLIP focusing on the Left/Right subsets of What'sUp, COCO-spatial, and GQA-spatial benchmark with or without substituted text encoder, after finetuning on LLaVA-1.5's training data with or without hard negative captions. After finetuning, the accuracies are still around or below random chance.

| | What'sUp Subset A | | What'sUp Subset B | | COCO-spatial | | GQA-spatial | |
| | Indiv. | Pairs | Indiv. | Pairs | One-obj. | Two-obj. | One-obj. | Two-obj. |
|---|---|---|---|---|---|---|---|---|
| CLIP-ViT-L/14-336px | 49.0 | 1.9 | 54.9 | 10.8 | 51.6 | 48.4 | 52.1 | 50.4 |
| + finetuning | 50.5 | 2.0 | 53.9 | 5.9 | 49.9 | 53.4 | 49.1 | 53.8 |
| + neg. cap. | 50.5 | 1.0 | 50.5 | 1.0 | 48.5 | 55.6 | 49.4 | 50.4 |
| + llm2vec, finetuning | 50.0 | 1.0 | 49.5 | 0.0 | 48.5 | 48.0 | 50.0 | 53.8 |
| + llm2vec, neg. cap. | 49.5 | 2.9 | 50.5 | 6.9 | 48.8 | 46.2 | 48.1 | 51.9 |
| SigLIP-ViT-L/16-384px | 50.0 | 1.9 | 51.5 | 5.9 | 48.7 | 50.2 | 51.2 | 47.0 |
| + finetuning | 49.0 | 1.0 | 51.0 | 3.9 | 50.8 | 53.1 | 49.7 | 55.3 |
| + neg. cap. | 50.0 | 0.0 | 50.0 | 0.0 | 50.5 | 53.8 | 51.0 | 48.1 |
| + llm2vec, finetuning | 50.5 | 2.9 | 51.0 | 3.9 | 50.1 | 54.8 | 49.1 | 51.1 |
| + llm2vec, neg. cap. | 50.5 | 1.0 | 51.0 | 3.9 | 50.0 | 47.7 | 48.8 | 50.0 |
| Random chance | 50.0 | 25.0 | 50.0 | 25.0 | 50.0 | 50.0 | 50.0 | 50.0 |

results in Appendix B.3). All the above results indicate that data alone are not to blame for the failure of CLIP-like models.

## 4.3 LANGUAGE MODEL CHOICES

Previous research suggested that the CLIP text encoder is "blind" too (Tong et al., 2024b; Kamath et al., 2023b; Yuksekgonul et al., 2023)– It struggles with capturing changed word orders, negation, and spatial or numerical details. On the other hand, LLaVA-1.5-7B employs a pretrained large language model (LLM), Vicuna-1.5-7B, which is supposed to be better than the CLIP text encoder at language reasoning, and the commonsense knowledge it learned during language modeling should benefit VLMs.

Is pretrained LLM the secret to the success of LLaVA-1.5? To answer this question, we perform further experiments on finetuning CLIP-like models using both the LLaVA-1.5 training data and a stronger text encoder. Since Vicuna-1.5-7B is a decoder-only language model, we utilize the LLaMA-2-7B-chat-hf-mntp checkpoint provided in Llm2vec (BehnamGhader et al., 2024), where LLaMA-2-7B-chat model was converted to a text encoder and showed excellent performance in encoding texts. We replace the original CLIP text encoder with this pretrained model and add a two-layer MLP connector on top of it to align its output dimension with the CLIP image encoder. Since this text encoder is well-trained, we freeze it during finetuning. To make a fair comparison, we lock the image encoder again and use the same connector design as the one used in LLaVA-1.5-7B with only differing widths.

We train the models in two settings: plain data and data with hard negative captions. The hard negative captions are constructed in the same way as in Section 4.2. The results are shown in Table 5. Surprisingly, a strong text encoder does not help either. Like the practice in Section 4.2, we try unlocking the image encoder in this setting (See Appendix B.3). We only observe a significant increase in the pair accuracy on What'sUp when a strong text encoder, hard negative captions, and unlocked image encoder are all used. Still, in this case, the individual accuracy and the accura-

Table 6: Results of LLaVA-1.5-7B with M3ID ($\alpha = 0.6, \lambda = 0.15$) using original evaluation on MMVP benchmark, along with the results of other methods.

|  | Indiv. Acc. | Pairs Acc. |
| --- | --- | --- |
| LLaVA-1.5-7B | 61.7 | 25.3 |
| w/ RP (Jiao et al., 2024) | – | 27.3 |
| w/ M3ID (Favero et al., 2024) | **64.3** | **31.3** |
| w/ DIVA (Wang et al., 2024a) | – | **31.3** |
| Random chance | 50.0 | 25.0 |

cies for the COCO-spatial and GQA-spatial do not improve. Based on these results, we argue that higher quality and relevant training data or stronger language models do not solely contribute to the performance gap.

By controlling other factors, we suggest that differences in VLM paradigms may largely explain the performance gap. One hypothesis is that CLIP-like models use a dot product for image-text alignment during training and inference. For a given image embedding, every text embedding is linearly projected into a spectrum $[-1, 1]$ regarding their matching degree. While this multimodal contrastive paradigm achieves great success in tasks like zero-shot classification, the learned alignment might not effectively capture all correspondences between image and text for various downstream tasks. This hypothesis aligns with our analysis in Section 3.1 that different visual information extraction strategies matter.

## 5 DISCUSSION

Although we find that visual information extraction methods matter a lot, and LLaVA-1.5 has a stronger extraction ability on highly similar embeddings, its poor performance on the MMVP benchmark remains unexplained (Tong et al., 2024c). In this section, we reconsider its failure and provide insight into future improvements based on two findings in MLLMs: They might not attend enough to the visual input, and the visual information is often aligned correctly but probably did not induce enough differences in the output token probability.

### 5.1 ALTERNATIVE DECODING FOR LLAVA

Inspired by the findings in Section 4.1 that MLLMs might "forget" the visual input gradually, one possible improvement is to "remind" MLLMs of them, magnifying the effect of visual input on language models. Multi-Modal Mutual-Information Decoding (M3ID) was designed for this purpose on MLLMs like LLaVA (Favero et al., 2024). For token in each decoding step $t$, M3ID computes the output probability with the image and without any input image, denoted as $\mathbf{l_c}$ and $\mathbf{l_u}$, respectively. The latter corresponds to the language prior. Then a correction term $(\mathbf{l_c} - \mathbf{l_u})$ is added to $\mathbf{l_c}$ with weight $\frac{1-\exp(-\lambda t)}{\exp(-\lambda t)}$ if the model is not highly confident with the next token $(\max_k(l_c)_k < \log \alpha)$. This correction prevents the VLM from omitting the visual input and relying on the language prior.

We test this decoding strategy on the MMVP benchmark in the standard setting. In Table 6, this method achieves the most gain (+6%) relative to the baseline LLaVA-1.5-7b. We note that this surpasses some methods that modified the vision part, such as Libra (30.0 with a decoupled and more complex vision system) (Xu et al., 2024) and is on par with I-MoF (31.3 with interleaved CLIP and DINO features) (Tong et al., 2024c). This result suggests that LLaVA-1.5 did not attend to the visual input enough and thus might miss the key information for answering the query. A similar finding was described through the interpretability perspective in Stan et al. (2024).

### 5.2 EVALUATION WITH RELAXED CONSTRAINTS

We look into the results of the MC evaluation and find that the output token probability often differs for two images (e.g., compared with image 2, image 1 slightly prefers caption 1 more). Still, the evaluation omits them since we always pick the caption with a higher probability for each im-

Table 7: Results of CLIP-ViT-L/14-336px and LLaVA-1.5-7B using original pair evaluation and new evaluation with relaxed constraints on MMVP benchmark.

|  | Original Evaluation | w/ Relaxed Constraints |
|---|---|---|
| CLIP-ViT-L/14-336px | 14.0 | 64.0 |
| LLaVA-1.5-7B | 25.3 | 73.3 |
| Random chance | 25.0 | 50.0 |

age. Such differences show that the visual nuances are often extracted and aligned with the correct semantics by LLaVA-1.5 rather than being discarded after visual encoding.

How many visual nuances are preserved and extracted by LLaVA-1.5? We explore this question by testing the LLaVA-1.5 on a new evaluation pipeline with relaxed constraints. To catch the slight difference in model output, similar to the MC evaluation, we calculate the model perplexity of two possible options. MC only uses the letters "A" and "B" when computing perplexity, but we use the full option for perplexity computation, e.g., "(a) Open" and "(b) Closed" in the questions provided by the original benchmark. Denote the perplexity of two options (normalized by the number of tokens) given two images to be $ppl_{i1c1}, ppl_{i1c2}, ppl_{i2c1}, ppl_{i2c2}$, respectively. We consider the model to be correct for this test case if they satisfy

$$\frac{ppl_{i1c1}}{ppl_{i1c1} + ppl_{i1c2}} > \frac{ppl_{i2c1}}{ppl_{i2c1} + ppl_{i2c2}},$$

In other words, the model considers (image 1, caption 1) with (image 2, caption 2) more possible matches than (image 2, caption 1) with (image1, caption 2). This way, we "force" the model to output differently for two images in a pair, and thus, the random chance is 50%. Through this comparison, we amplify the semantics induced by visual nuances. We also apply this evaluation on CLIP, replacing perplexity with cosine similarity. In Table 7, the new performance is significantly higher than random chance as the baseline (+23.3%), compared with the pair accuracy under the original evaluation (+0.3%). This means more visual information can be extracted from the image embedding and aligned with the correct semantics than the original results suggested.

The possible reason why they failed to be extracted in the original setting is that the language model did not fully utilize the image-induced semantics. Consequently, they failed to affect the output probability enough to produce the correct answer. Influenced by language prior and spurious correlation with irrelevant text tokens (Ye et al., 2024), it will probably output common answers or even hallucinations. Hence, reaching this "upper bound" in the original evaluation requires the VLM to utilize extracted visual nuances properly and balance it with language prior during generation.

## 6 CONCLUSION

Our study questions the use of erroneous agreements to reflect CLIP image encoders' information loss or blindness and finds that they are not the sole cause of VLM failures. We show that the amount of extracted visual information largely depends on the extraction strategy, which varies widely across VLMs. LLaVA-1.5, with a stronger extraction ability, outperforms CLIP-like models on our benchmarks. Our controlled experiments suggest that the key factor in their performance discrepancy might lie in their paradigms. We believe the information loss of the image encoder should be defined when conditioning on the VLM paradigm and possibly the downstream task.

Our results suggest there is still room to enhance VLMs with a fixed, pretrained image encoder. While balancing the visual grounding ability and image-text correspondence (e.g., combining popular visual representation learning models in various styles) could reach the best trade-off on the curve for heterogeneous benchmarks, developing advanced methods for VLMs to extract and utilize given visual information might shift the curve upwards.

**Limitation.** We view the VLMs abstractly and do not look into fine-grained details on how the visual information is extracted and leads to the model's output. We leave its dissection for future research. For ablation studies, we do not train CLIP or SigLIP models from scratch or use larger batch sizes due to the limitation in computing resources, so the conclusion on the effects of different factors is restricted.

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

Table 8: Question formats for different subsets.

| Subset | Question |
|---|---|
| What'sUp Subset A&B, Left/Right | Is the (object 1) to the left of or to the right of the (object 2)? Answer left or right. |
| What'sUp Subset A, On/Under | Is the (object 1) on or under the (object 2)? Choose from the two options. |
| What'sUp Subset B, Front/Behind | Is the (object 1) in front of or behind the (object 2)? Answer front or behind. |
| COCO/GQA-spatial, One obj. | Is the (object 1) on the (left/right/top/bottom) or on the (right/left/bottom/top)? Give a short answer. |
| COCO-spatial, Two obj. | Is the (object 1) (to the left of/to the right of/above/below) a (object 2) or (to the right of/to the left of/below/above) a (object 2)? Give a short answer. |
| GQA-spatial, Two obj. | Is the (object 1) to the (left/right/front/behind) of a (object 2) or to the (right/left/behind/front) of a (object 2)? Give a short answer. |

## A    BENCHMARKS AND EVALUATIONS

We use the public pretrained weights of LLaVA-1.5-7B (`https://huggingface.co/llava-hf/llava-1.5-7b-hf`) for evaluation and use greedy encoding by default to ensure reproducibility. We use OpenAI's pretrained CLIP-ViT-L/14-336px model, SigLIP-ViT-L/16-384px pretrained on the WebLI dataset (Chen et al., 2022) provided in the OpenCLIP repository, and official pretrained XVLM-16M weight for both evaluation and finetuning.

### A.1    EVALUATION ON WHAT'SUP

The What'sUp benchmark (Kamath et al., 2023a) contains 820 images of pairs of household objects, 408 in Subset A and 412 in Subset B. We corrected the mislabeled images in the GitHub Issues and reevaluated the pretrained VLMs. For CLIP, SigLIP, and XVLM's evaluation, we use the official code provided by the What'sUp benchmark's authors in `https://github.com/amitakamath/whatsup_vlms`.

For LLaVA-1.5, the questions used for evaluation are listed in Table 8. Then the question is concatenated with the fixed prompt template ("USER: <image> \n(question) ASSISTANT:"). Considering the position bias in LLMs (Wang et al., 2024b), we exchange the position of two prepositions in the question with 50% probability on COCO-spatial and GQA-spatial benchmarks for fair results. On the What'sUp benchmark, the orders are always the same for two images. Then, we evaluate the outputs by keyword matching since we observe that the output is quite structured.

The reason why we use different commands after the main question (e.g., "Answer left or right", "Choose from the two options", and "Give a short answer") is that we found the LLaVA-1.5 model sensitive to such command. We tried "Answer on or under" for the On/Under subset in What'sUp Subset A, and the model accuracy is quite low. This is one of its limitations that deserves future research. However, we aim to show that LLaVA-1.5 can extract such information, so we use the best prompt to showcase its ability.

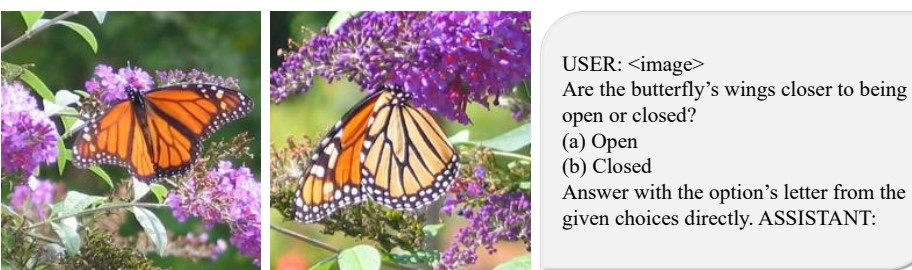

Figure 4: Example test case and prompt for LLaVA-1.5 in MMVP benchmark.

## A.2 Evaluation on MMVP and MMVP-VLM

The MMVP benchmark contains 150 pairs of similar images, and the MMVP-VLM benchmark has 135 pairs of similar images, divided into nine categories. There is an overlap between the image pairs in these two benchmarks. An example of an image pair and the corresponding prompt for LLaVA-1.5 in MMVP are shown in Figure 4. We corrected the mislabeled images in the GitHub Issues and reevaluated the pretrained VLMs. Since MMVP is incompatible with CLIP and so is MMVP-VLM with MLLMs, we convert their questions manually. We attach these new questions to the supplementary material for reference.

In the standard setting, we evaluate the correctness of the model response by human evaluation. Although the accuracy given by the GPT-4 evaluation was close to that of a human evaluation on average, we noticed that it is unreliable since it gave several wrong judgments. So, we evaluate the correctness of the answers manually to avoid models getting higher accuracy by cheating GPT-4. In the Multiple-Choice setting, we calculate and rank the perplexity of "A" and "B" given by the model.

In Section 5.2, we calculate the perplexity of two options (The two options are "(a) Open" and "(b) Closed" for the example in Figure 4). We also add the EOS ("") to the end of these options and normalize the perplexity by their number of tokens.

# B Supplementary Experimental Details and Results

## B.1 Hyperparameters

Our code is based on https://github.com/mlfoundations/open_clip. We finetune CLIP and SigLIP models for 5 epochs with a learning rate of $5e-6$ on the combination of converted LCS-558K plus converted DataMix-665K. We use 50 steps of warmup and AdamW optimizer with a cosine-annealing learning rate schedule. The batch size is 512, and we train the models on 4 gpus.

## B.2 LLaVA-1.5's Training Data

We check the frequency of appearance of the following keywords in DataMix-665K and LCS-558K: "on the left," "on the right," "to the left," "to the right," "at the left," "at the right." In DataMix-665K, there are 12957 instances with at least one of the key phrases, among which 12658 have a paired image. For captions (ground truth answers), this number is 13473 since an instance is paired with a multi-turn conversation. In LCS-558K, there are 560 such instances and captions since each instance has only one question and one answer.

In our experiments in Section 4.2, LCS-558K was converted from image-text pair format to conversation format, so we revert this process by using ground truth answer as the caption. Since DataMix-665K is in a multi-turn conversation format, we randomly pick one answer as the caption in each epoch. In Section 4.3, the new text encoder can encode long paragraphs, so we use the concatenation of all answers in the multi-turn conversation as the ground truth caption.

Table 9: Results of SigLIP-ViT-L/16-384px on the Left/Right subsets of What'sUp, COCO-spatial, and GQA-spatial benchmark after finetuning with image encoder unlocked on LLaVA-1.5's training data (LCS-558K + DataMix-665K) with constructed hard negative captions.

| | What'sUp Subset A | | What'sUp Subset B | | COCO-spatial | | GQA-spatial | |
| --- | --- | --- | --- | --- | --- | --- | --- | --- |
| | Indiv. | Pairs | Indiv. | Pairs | One-obj. | Two-obj. | One-obj. | Two-obj. |
| SigLIP-ViT-L/16-384px | 50.0 | 1.9 | 51.5 | 5.9 | 48.7 | 50.2 | 51.2 | 47.0 |
| + finetuning | 50.5 | 2.9 | 51.5 | 5.9 | 48.7 | 57.7 | 50.5 | 48.1 |
| + neg. cap. | 50.0 | 3.9 | 47.1 | 2.0 | 52.3 | 47.0 | 51.8 | 52.7 |
| Random chance | 50.0 | 25.0 | 50.0 | 25.0 | 50.0 | 50.0 | 50.0 | 50.0 |

Table 10: Results of SigLIP-ViT-L/16-384px on the Left/Right subsets of What'sUp, COCO-spatial, and GQA-spatial benchmark. We substituted the text encoder to be Llama-2-7b-chat-hf-mntp, then finetuned the model with image encoder unlocked on LLaVA-1.5's training data (LCS-558K + DataMix-665K) with or without constructed hard negative captions.

| | What'sUp Subset A | | What'sUp Subset B | | COCO-spatial | | GQA-spatial | |
| --- | --- | --- | --- | --- | --- | --- | --- | --- |
| | Indiv. | Pairs | Indiv. | Pairs | One-obj. | Two-obj. | One-obj. | Two-obj. |
| SigLIP-ViT-L/16-384px | 50.0 | 1.9 | 51.5 | 5.9 | 48.7 | 50.2 | 51.2 | 47.0 |
| + finetuning | 50.0 | 1.0 | 49.0 | 7.8 | 50.9 | 50.2 | 48.7 | 50.8 |
| + neg. cap. | 56.3 | 26.2 | 55.4 | 25.5 | 50.8 | 48.4 | 46.7 | 53.4 |
| Random chance | 50.0 | 25.0 | 50.0 | 25.0 | 50.0 | 50.0 | 50.0 | 50.0 |

### B.3 RESULTS OF UNLOCKING IMAGE ENCODER

We try unlocking the image encoder during finetuning on the SigLIP model. The results after finetuning with CLIP text encoder are in Table 9, and results with LLaMA-2-7B-chat-hf-mntp are in Table 10. Interestingly, we observe a significant increase in pair accuracy on the What'sUp benchmark only when using hard negative captions and a strong text encoder while unlocking the image encoder. Still, the individual accuracy remains low.

### B.4 RESULTS OF FINETUNING XVLM

Observing the similar failure of the data-informed attempt, previous work concluded that even with relevant, high-quality data and hard negatives, denser supervision is likely required to let the model learn the basic spatial relations (Kamath et al., 2023a), as in XVLM (Zeng et al., 2021), a VLM with supervision at the bounding-box level. However, LLaVA does not incorporate downstream task-related inductive bias or denser supervision to achieve high accuracy, yet it beats XVLM finetuned on COCO (Lin et al., 2014) on the What'sUp benchmark.

We explore finetuning XVLM on LLaVA's training data based on their official code (https://github.com/zengyan-97/X-VLM), but no improvement is observed in the results (the last two model rows in Table 11). The image encoder is locked during finetuning. We use both contrastive learning loss and image-text matching loss. The evaluation is performed through the image-text matching score. We finetune the XVLM-16M model for 5 epochs with a learning rate of $1e-5$ and a weight decay rate of $0.01$. We use 10% steps of warmup and AdamW optimizer with a lambda learning rate schedule. The batch size is 128, and we train the model on 4 gpus.

### B.5 RESULTS OF DIFFERENT MLLMS

Do our findings on LLaVA-1.5 in Section 3.2 generalize to other MLLMs? We evaluate four other MLLMs using their officially released weights. First, we consider LLaMA-3-V-8B and Phi-3-V-3.8B which have LLaVA-like architecture and use frozen CLIP-ViT-L/14-336px as the image encoder (Rasheed et al., 2024). For MLLMs with different architectures and training data, we use Otter-Image-MPT7B (Li et al., 2023b;a) with frozen CLIP-ViT-L/14 as the image encoder and

Table 11: Results of XVLM-16M on the Left/Right subsets of What'sUp, COCO-spatial, and GQA-spatial benchmark on LLaVA-1.5's training data.

| | What'sUp Subset A | | | What'sUp Subset B | | | COCO-spatial | | GQA-spatial | |
| | Indiv. | Pairs | Set of 4 | Indiv. | Pairs | Set of 4 | One-obj. | Two-obj. | One-obj. | Two-obj. |
| --- | --- | --- | --- | --- | --- | --- | --- | --- | --- | --- |
| XVLM-16M | 50.0 | 30.6 | 1.0 | 32.8 | 10.8 | 0.0 | 65.4 | 64.6 | 63.2 | 53.3 |
| + finetuning | 46.4 | 28.4 | 1.0 | 34.6 | 8.3 | 1.0 | 66.8 | 65.2 | 61.3 | 51.2 |
| Random chance | 25.0 | 6.3 | 0.4 | 25.0 | 6.3 | 0.4 | 50.0 | 50.0 | 50.0 | 50.0 |

Table 12: Results of CLIP-ViT-L/14-336px and MLLMs on four subsets in What'sUp. The individual accuracy and pair accuracy are in percentage points. The average cosine similarity of the CLIP-ViT-L/14-336px image embeddings for image pairs is calculated for each category.

| | What'sUp Subset A | | | | What'sUp Subset B | | | |
| | Left/Right | | On/Under | | Left/Right | | Front/Behind | |
| | Indiv. | Pairs | Indiv. | Pairs | Indiv. | Pairs | Indiv. | Pairs |
| --- | --- | --- | --- | --- | --- | --- | --- | --- |
| CLIP-ViT-L/14-336px | 49.0 | 1.9 | 61.7 | 23.3 | 54.9 | 10.8 | 51.5 | 7.8 |
| LLaVA-1.5-7B | 99.0 | 98.1 | 80.1 | 60.2 | **100** | **100** | **98.5** | **97.1** |
| LLaMA-3-V-8B | 90.3 | 80.6 | 57.8 | 20.4 | 71.1 | 46.1 | 69.1 | 41.2 |
| Phi-3-V-3.8B | **100** | **100** | 85.4 | 70.9 | **100** | **100** | 56.9 | 13.7 |
| InstructBLIP-Vicuna-7B | 50.0 | 1.9 | **93.7** | **87.4** | 50.0 | 0.0 | 50.0 | 5.9 |
| Otter-Image-MPT7B | 50.0 | 1.0 | 56.8 | 13.6 | 50.0 | 0.0 | 51.5 | 11.8 |
| Avg. Embedding Cosine Sim. | 0.995 | | 0.971 | | 0.955 | | 0.902 | |

InstructBLIP-Vicuna-7B (Dai et al., 2023) with frozen EVA-CLIP-ViT-G/14 (Sun et al., 2023). Otter adopts the OpenFlamingo (Awadalla et al., 2023) paradigm with a Perceiver resampler module on top of the frozen image encoder, and then sends the output of this module to the cross-attention layers of the language model. InstructBLIP employs the pretrained BLIP-2 (Li et al., 2023c) model, with a Q-Former and a fully connected layer as the vision-language connector between the frozen image encoder and the language model. Inside the Q-Former, image embeddings are used in cross-attention layers.

During evaluation, we find that all of these MLLMs are sensitive to the wording in the command part, so we try several commands and report the best results as we did for LLaVA-1.5. We keep all other settings the same as in Section A.1.

The results are shown in Table 12 and Table 13. For comparison, we also include the results of CLIP-ViT-L/14-336px and LLaVA-1.5. The good performance of LLaMA-3-V-8B and Phi-3-V-3.8B verifies that they can also extract distinct information from highly similar embeddings, though they are weak on some prepositions (Front/Behind for Phi-3-V-3.8B, and On/Under for LLaMA-3-V-8B). These results show that our findings generalize to these two MLLMs with different scales and language models.

On the other hand, with different architectures and training data, Otter and InstructBLIP still struggle on this benchmark (except On/Under for InstructBLIP). Hence, MLLMs do not guarantee effective extraction from frozen image encoder. Good design of MLLM architecture and curated training data synergize to provide strong visual information extraction ability.

Table 13: Results of CLIP-ViT-L/14-336px and MLLMs on What'sUp (four-way classification) benchmark, COCO-spatial, and GQA-spatial. "Set of 4" is the correctness for all four images in a set.

| | What'sUp Subset A | | | What'sUp Subset B | | | COCO-spatial | | GQA-spatial | |
| --- | --- | --- | --- | --- | --- | --- | --- | --- | --- | --- |
| | Indiv. | Pairs | Set of 4 | Indiv. | Pairs | Set of 4 | One-obj. | Two-obj. | One-obj. | Two-obj. |
| CLIP-ViT-L/14-336px | 28.9 | 1.0 | 0.0 | 27.2 | 1.0 | 0.0 | 48.9 | 51.1 | 46.6 | 49.1 |
| LLaVA-1.5-7B | **62.1** | **41.3** | 14.6 | **74.0** | **61.8** | **23.5** | 96.0 | 82.3 | 96.0 | 90.7 |
| LLaMA-3-V-8B | 60.0 | 36.4 | **17.5** | 70.1 | 43.6 | 20.6 | **97.8** | 83.2 | **99.0** | 90.7 |
| Phi-3-V-3.8B | 58.0 | 36.4 | 15.5 | 71.8 | 55.4 | 12.8 | 97.3 | **85.2** | 98.0 | **91.1** |
| InstructBLIP-Vicuna-7B | 37.6 | 25.7 | 0.0 | 29.9 | 15.2 | 0.0 | 55.0 | 51.4 | 47.8 | 50.2 |
| Otter-Image-MPT7B | 24.5 | 2.4 | 0.0 | 24.8 | 3.0 | 0.0 | 51.9 | 50.0 | 54.1 | 51.9 |
| Random chance | 25.0 | 6.3 | 0.4 | 25.0 | 6.3 | 0.4 | 50.0 | 50.0 | 50.0 | 50.0 |

