# OpenReview forum: "On Erroneous Agreements of CLIP Image Embeddings"
_ICLR.cc/2025/Conference — Submitted to ICLR 2025_

### Official Review · Reviewer_7vBu · 2024-10-28

**Soundness:** 3
**Presentation:** 3
**Contribution:** 3
**Rating:** 5
**Confidence:** 4

**Summary:**

Previous works have argued that the poor performance of VLMs on simple visual reasoning tasks is due to their dependence on CLIP encoder. They show that CLIP can encode two visually different images with high cosine similarity (called erroneous agreement) and argue that many VLMs fail due because they use CLIP as their vision encoder.

In this paper the authors show that with better extraction and utilization methods, clip encoder can still be used for downstream tasks of visual reasoning. They show experiments with LLaVA-1.5 and show that it performs good on benchmarks despite using CLIP as its vision encoder.

**Strengths:**

1. Important analysis shown in section 4 (Investigating the performance gap)- this section answers the questions related to training data, language model and evaluation method. This analysis is important to make the claim that visual information extraction is the key factor in determining the performance gap on downstream tasks. And these other factors (eval method, language encoder, training data) are not contributing much to the improved performance.

2. Detailed benchmarking of the models on different datasets and good ablation studies.

3. They show, using a different decoding method, that even with a fixed pre-trained image encoder if we try to 'force' VLMs to attend to visual features while decoding (and not just relying on language priors), we can perform good on downstream visual reasoning tasks.  Although they used a previously proposed decoding strategy M3ID  (Favero et al., 2024).

**Weaknesses:**

1. The authors show that the visual feature extraction technique in LLaVA (a two layer MLP) is an important step in distinguishing between two erroneous images. But they do not provide an convincing argument on why is it an important step. An analysis on "why just adding a 2-layer MLP on top of pre-trained CLIP makes it so much better?" would have been an amazing addition to the paper.

2. On Spearman's rank correlation (also asked in the questions): Since CLIP is trained using loss based on cosine similarity, I think using Spearman's rank correlation to show that two embeddings are "fully opposed" is not correct. For example, consider the example given on LN 232-233. Although the ranks of the dims are reversed giving  ρ = −1, their absolute values are pretty close. And if we assume (in an ideal world) them to be separable features, for example the embeddings could be of dog images and the features are 'ear-length' , 'fur color', 'nose-shape', both the embeddings will still show two very similar looking dogs (and not 'fully opposite') even though the embedding might have ρ = −1.

**Questions:**

Would a high negative Spearman's rank correlation show that the embeddings are quite different?

LN 232-236 says: "While SC (fv(v1), fv(v2)) > 0.989, Spearman’s rank correlation coefficient can tell their sharp difference: ρ = −1, showing that they are fully opposed in this sense. Therefore, the difference in visual inputs might still be extracted through other means when erroneous agreements occur"

How does ρ = −1 show that the embeddings are 'fully opposed'? If the authors could show this or cite a paper that shows this, that would be great.

---

> ### Author Response · Authors · 2024-11-22
> **Our Response to Your Concerns**
>
> Thank you for your detailed comments on our paper! We hope that our response below will help you better understand our paper.
>
> 1. **Your concerns about not showing that a two-layer MLP is important in distinguishing two erroneously agreeing images.**
>
>     We **did not suggest that the two-layer MLP is the sole key factor** in the performance gap. The argument in our paper is that the visual information extraction strategies of VLMs, by which we mean techniques applied on top of the image encoder, are important. In LLaVA-1.5, this refers to both the two-layer MLP connector and the language model used for answer generation.
>
>     Although separating the functions of the connector and the language model in MLLMs is an interesting question for future exploration, this is not the focus of our paper. To avoid future confusion, we clarified the meaning of visual information extraction in the introduction of our revised submission (L74-L76).
> 2. **Your concerns regarding our use of Spearman's rank correlation and why ρ = −1 show that the embeddings are "fully opposed."**
>
>     In our paper, Spearman's rank correlation coefficient serves as a toy example to show that cosine similarity does not depict all aspects of vector pairs, and we could still get different information from vectors with high cosine similarity.
>
>     For the vector pairs in our example ($[10,11,12]^\top$ and $[12,11,10]^\top$), their Spearman's rank correlation coefficient is -1, indicating that they have a perfectly inverse order. Hence, their order information is fully opposed in this sense.
>
>     For your example of interpreting these two vectors as the embeddings of two dog images, we did not intend to show that the dogs "look opposite in the general sense." Instead, we show that in the specific sense (order information), the two embeddings are opposite, and we can extract this opposite information through Spearman's rank correlation coefficient. If they are indices of the dog's features you mentioned, the two embeddings could yield opposite answers to the question, "Are the indices of the dog's features in an ascending order?"
>
>     We also rephrased the relevant description (L243-L244) for the toy example in our submission to better express our intention.
>
> Feel free to ask further questions, and we are willing to address them as well!

---

> > ### Comment · Reviewer_7vBu · 2024-11-26
> >
> > I thank the authors for their response. I would like to stay with my rating.

---

> > > ### Author Response · Authors · 2024-11-27
> > >
> > > We thank you for your reply and for taking the time to consider our responses. If there are any points where the paper could benefit from additional clarification or revisions, we would be happy to address them. If there are no further concerns and you find the revised version satisfactory, we kindly ask if you consider revisiting your score. Again, we value your review and hope our revision align with your expectations.
> > >
> > > Thank you again for your time and constructive engagement.

---

### Official Review · Reviewer_DMVn · 2024-10-30

**Soundness:** 2
**Presentation:** 3
**Contribution:** 2
**Rating:** 5
**Confidence:** 3

**Summary:**

This paper examines the performance of LLaVA-1.5-7B on visual reasoning tasks, specifically WhatsUp and MMVP, and concludes that its suboptimal performance is not due to CLIP's visual features. While CLIP visual features effectively capture semantic similarities, they occasionally misinterpret spatial differences in object placement (e.g., "mug on the left" vs. "mug on the right"), which results in high cosine similarity (over 0.95) despite subtle image differences—referred to as "erroneous agreements." The authors show that CLIP’s visual features are accurate; instead, they attribute the performance issues to LLaVA not making effective use of these features. They further demonstrate that poor alignment between visual and textual inputs, not the visual features themselves, explains the bad performance in CLIP models for these tasks and datasets. Unlike CLIP, LLaVA does not exhibit this alignment problem, and this is shown quantitatively. Finally, the authors try better decoding strategies in Llava like M3ID such that the decoding better makes use of the visual features. They also show that  multiple image inputs works better to highlight the difference in images. They also explore performance gaps related to evaluation methods, training data, and the text encoder.

**Strengths:**

- This paper delivers a valuable message to the community by advocating for enhancing Multimodal LLMs and keeping the image encoder fixed. Previous research suggested that the image encoder introduced issues by producing "erroneous agreements" (similar embeddings for semantically similar but visually distinct images). However, this paper counters that claim, attributing the problem instead to the the model not utilizing these visual features effectively.

- Interesting observation of better decoding algorithms and methods for evaluating specific tasks.

**Weaknesses:**

- There is an incoherent story. The abstract initially suggests that LLaVA performs well on reasoning tasks and achieves high accuracy, yet later the paper claims LLaVA performs poorly on MMVP, contradicting the initial statement. They also mention that LLava is able to extract the correct information from the visual features, and that it does not face issues (L186, and demo image). Only later is it clarified that LLaVA performs well on WhatsUp but not on MMVP. In general, I feel there is an unclear and confusing story.

- WhatUp, MMVP, COCO-spatial and GQA-spatial are not really well-known datasets and publicly-agreed on to measure reasoning. I actually came to know them after reading this paper. Measuring reasoning on MMLMs are usually not done on these datasets. These datasets are not enough to reflect model reasoning and to come up with general conclusions about LLava or MMLMs in general. The authors don’t show ablation and analysis results using their ablation strategies, on important reasoning tasks such as VQA, GQA, OK-VQA, VCR and others (specifically, those that LLava reports on). I feel the scope, task and datasets are not enough to reach the standard required for ICLR.

**Questions:**

In general the second weakness is the biggest to me. I would like to hear what the authors say on this?

---

> ### Author Response · Authors · 2024-11-22
> **Our Response to Your Concerns**
>
> Thank you so much for your detailed feedback! Below, we offer responses to assure your concerns:
> 1. **Your feedback that the general storyline of our paper is confusing.**
>
>     Thank you for your valuable suggestion in our writing!
>
>     Our storyline is as follows: Previous research attributes LLaVA-1.5's low performance on the MMVP benchmark to deficiencies in the CLIP vision encoder, specifically erroneous agreements. We first show that LLaVA-1.5 is able to perform well on image pairs with erroneous agreements as shown on the What'sUp benchmark, whereas CLIP cannot. Then we conduct an ablation study to identify the key difference between CLIP and LLaVA-1.5 that causes their performance gap. Finally, we revisit the MMVP benchmark and provide insights into the true cause for LLaVA-1.5's shortcomings.
>
>     In our paper, What'sUp is considered to be easier for LLaVA-1.5 than MMVP, and the latter remains unsolved. Since the What'sUp and MMVP benchmarks are less known than the benchmarks you mentioned, it is useful to introduce and explain more on model performance and what people expected on them previously. In our submission, we introduced this difference in the introduction (L86) and in task setup (L223). Now we added more context to Figure 1 (both the image and the caption), highlighted this difference more in the introduction (L52-L54, L67), and revised the relevant description (originally in L186, now in L193-L195) to improve reader experience.
>
> 2. **Your concerns that our scope, task, and dataset choices are not enough to derive general conclusion on the visual reasoning abilities of LLaVA and MMLMs.**
>
>     Our goal is not to draw conclusions about the **general visual reasoning** abilities of LLaVA or other MMLMs or to prove their supremacy over CLIP generally. We focus on the specific type of task that requires the VLMs to distinguish image pairs with highly similar CLIP embeddings (erroneous agreements), which were claimed to cause VLMs' shortcomings in the paper "Eyes Wide Shut? Exploring the Visual Shortcomings of Multimodal LLMs." This motivates our use of the What'sUp and MMVP benchmarks for comparisons and ablation studies, as they consist of such image pairs. On these benchmarks, we show contrary evidence that LLaVA-1.5 can extract distinct information even from these similar embeddings.
>
>     The other benchmarks you mentioned, including VQA, GQA, OK-VQA, VCR, and those used by LLaVA (VisWiz, SQA, TextVQA, POPE, MME, MMB, Chinese MMB, SEED, LLaVA-Bench, MM-Vet), are valuable for evaluating general visual reasoning capabilities. However, since they do not focus on distinguishing similar paired images, they fall outside the scope of our study.
>
>     Although our scope is limited to this specific task type, our findings highlight important insights: CLIP and LLaVA-1.5 paradigms employ inherently different mechanisms for extracting visual information, and there is still room to enhance VLMs with a fixed, pretrained image encoder. These findings contribute to the broader goal of designing more capable VLMs for general visual reasoning tasks.
>
>     We further clarified our scope in the related work section (L104-L107) and add more captions to Figure 2 to avoid confusion.
>
> If you have further concerns or questions, please let us know and we are willing to reply as well!

---

> > ### Comment · Reviewer_DMVn · 2024-11-23
> > **response to authors**
> >
> > I thank the authors for their response. I wish to stay with my score.

---

> > > ### Author Response · Authors · 2024-11-27
> > >
> > > Thank you for taking the time to review our responses. If there is anything else we can clarify or improve in the paper, we would be happy to address it. If you feel the revised version addresses your concerns, we kindly ask you to consider updating your score.
> > >
> > > We truly value your feedback and hope our revisions meet your expectations. Thank you again for your time and thoughtful engagement.

---

> > > > ### Comment · Reviewer_DMVn · 2024-11-28
> > > > **response**
> > > >
> > > > I thank the authors for the revised version. However, i would like to maintain my score, as I find the claim that the authors make not well-supported by experimental evidence. The benchmarks the authors used are not enough to support this claim. Specifically, I asked for experiments on other datasets, including the ones used in the original paper [R1] that the authors counteract, but the authors did not address this during the discussion period. Although I understand that WhatsUp and MMVP are the ones related to erroneous agreements, I still don't think there can be generalization insights by just investigating these datasets. Moreover, only LLava is considered, and we cannot get generalizable insights from one model. The results could be of the way Llava is trained or because of its architecture. More MLLMs are needed to verify this claim. Given these reasons, i maintain my score.
> > > >
> > > >
> > > > [R1] Eyes Wide Shut? Exploring the Visual Shortcomings of Multimodal LLMs

---

> ### Author Response · Authors · 2024-12-02
> **Our Response to Your Concerns**
>
> Thank you for your reply! Below, we offer further explanation and evidence to assure your concerns:
>
> 1. *"The benchmarks the authors used are not enough to support this claim. Specifically, I asked for experiments on other datasets, including the ones used in the original paper [R1] that the authors counteract, but the authors did not address this during the discussion period."*
>
>     The benchmarks used in the original paper [R1] include MMVP/MMVP-VLM, LLaVA-Bench, POPE, LLaVA-In-the-Wild, MMBench, TextVQA, VQA-v2, and MM-Vet. Apart from MMVP/MMVP-VLM we used, the rest do not focus on distinguishing two images with erroneous agreements, and thus do not have paired similar images with opposing answers to questions. Hence, results on these benchmark do not contribute to illustrate that LLaVA-1.5 shows a better visual information extraction ability than CLIP on images with erroneous agreements.
>
>     To address the need for more generalizable insights, we tested CLIP and LLaVA-1.5 on another benchmark, NaturalBench [R2], to show the generalizability of our argument. It is a vision-centric benchmark that challenges vision-language models with pairs of simple questions about natural imagery. This dataset contains 1,900 test cases, each consisting of two images and two questions with opposing answers. Since NaturalBench follows MLLM's format but not CLIP's, we converted the questions and options into captions using GPT-4o-mini for CLIP evaluation. We report the individual accuracy and pairs accuracy as we did in Table 3 of our submission. Below are the results:
>
>     *Table 1: Results of CLIP-ViT-L/14-336px and LLaVA-1.5-7B on NaturalBench. The results for LLaVA-1.5-7B come from the Acc. and Q-Acc. in Table 1 of the NaturalBench paper.*
>
>     | | Indiv. Acc.  | Pairs Acc |
>     | --- |:-----:|:----:|
>     | CLIP-ViT-L/14-336px | 56.5 |  20.5  |
>     | LLaVA-1.5-7B        | **67.3** |  **37.7**  |
>     | Random Chance       | 50.0 |  25.0  |
>
>      Given that the images were not collected by high CLIP cosine similarity, we further consider their performance on 320 testcases in NaturalBench with the highest CLIP image cosine similarity (>0.85):
>
>      *Table 2: Results of CLIP-ViT-L/14-336px and LLaVA-1.5-7B on 320 testcases from NaturalBench.*
>
>     | | Indiv. Acc.  | Pairs Acc |
>     | --- |:-----:|:----:|
>     | CLIP-ViT-L/14-336px | 53.7 |  13.9  |
>     | LLaVA-1.5-7B        | **63.0** |  **28.8**  |
>     | Random Chance       | 50.0 |  25.0  |
>
>     From the results, we can see that the performance gap between LLaVA-1.5 and CLIP on similar image pairs generalizes to this benchmark, which has larger size than MMVP and has more diverse visual content than What'sUp, showing the generalizability of our finding.
>
> 2. *"Moreover, only LLava is considered, and we cannot get generalizable insights from one model. The results could be of the way Llava is trained or because of its architecture."*
>
>     First, we showed by additional results of LLaMA-3-V-8B and Phi-3-V-3.8B that our findings generalize to other LLaVA-like models of varying scales and language backbones.
>
>     Second, we acknowledge the possibility that the way LLaVA-like models are trained or their architecture might be important to realize strong extraction ability. We did not intend to prove the advantage of all MLLMs over their image encoders, since we note that MLLMs adopt various model architectures and training pipelines, and some of them might not be effective in visual information extraction. Nevertheless, our findings on LLaVA-like models show the feasibility of extracting distinct information from highly similar image embeddings through MLLMs, together with the effectiveness of LLaVA's methodology compared with CLIP on such image pairs.
>
> We hope these additional analyses and clarifications address your concerns. Thank you for your valuable feedback!
>
> [R2] NaturalBench: Evaluating Vision-Language Models on Natural Adversarial Samples.

---

### Official Review · Reviewer_2b8n · 2024-11-02

**Soundness:** 3
**Presentation:** 3
**Contribution:** 2
**Rating:** 5
**Confidence:** 2

**Summary:**

The paper challenges the prevailing belief that Vision-Language Models' (VLMs) failures in visual reasoning are primarily due to CLIP image encoder's "erroneous agreements" (where distinct images have high cosine similarity). Using LLaVA-1.5-7B as an example, they demonstrate that MLLMs can successfully extract distinct information from similar image embeddings, achieving high accuracy on tasks where CLIP performs poorly. This suggests that the limitation lies not in the image embeddings themselves, but in how effectively models extract and utilize the encoded information.

**Strengths:**

Provides compelling empirical evidence through controlled experiments across multiple benchmarks.
Challenges and refines an important assumption in the field about VLM limitations.
Demonstrates that existing architectures might be more capable than previously thought, just requiring better utilization strategies.

**Weaknesses:**

The paper's scope might be too focused on LLaVA-1.5 as the primary example, potentially limiting the generalizability of findings
While the paper shows that information can be extracted from similar embeddings, it doesn't fully tackle why LLaVA-1.5 is able to do this.

**Questions:**

How do these findings generalize to other MLLMs beyond LLaVA-1.5?
What specific mechanisms allow MLLMs to extract distinct information from seemingly similar embeddings?

---

> ### Author Response · Authors · 2024-11-22
> **Our Response to Your Concerns**
>
> Thank you for your feedback on our paper and your insightful questions! We really appreciate your suggestions, which helps improve our work. Here are our responses to your questions:
> 1. **How do these findings generalize to other MLLMs beyond LLaVA-1.5?**
>
>     We first would like to clarify that we did not intend to show that all MLLMs are better than CLIP. Instead, our main focus is to explore whether erroneous agreements are bottlenecks for VLMs (whether we can extract distinct information from highly similar CLIP embeddings or not), and we found that LLaVA-1.5 can capture such nuance, while CLIP cannot.
>
>     Regarding your question, we agree that generalizing our finding to other MLLMs is an interesting direction and good supplementary to our work. For serving our purpose above, we focus on MLLMs that have CLIP as the image encoder and freeze it during training (otherwise, this variable is not controlled). Hence, we evaluate two other MLLMs, **LLaMA-3-V-8B** and **Phi-3-V-3.8B**, which use frozen CLIP-ViT-L/14-336px as the vision encoder. We use the model weights provided in https://github.com/mbzuai-oryx/LLaVA-pp.
>
>     *Table 1: Results of LLaMA-3-V-8B and Phi-3-V-3.8B on What'sUp Subset A. For comparison, we also include the results for CLIP-ViT-L/14-336px and LLaVA-1.5-7B from the Table 1 in our submission.*
>
>     |                     | Left/Right, Indiv. | Left/Right, Pairs | On/Under, Indiv. | On/Under, Pairs |
>     | -------- |:-----------------------------:|:-------------------------------:|:-----------------------------:|:-------------------------------:|
>     | CLIP-ViT-L/14-336px |           49.0             |             1.9             |            61.7            |            23.3           |
>     | LLaVA-1.5-7B        |        99.0             |             98.1            |            80.1            |            60.2           |
>     | LLaMA-3-V-8B        |   90.3             |             80.6            |            57.8            |            20.4           |
>     | Phi-3-V-3.8B        | **100**            |             **100**             |            **85.4**            |            **70.9**           |
>     | Random Chance       |          50.0          |          25.0          |          50.0         |          25.0         |
>
>     *Table 2: Results of LLaMA-3-V-8B and Phi-3-V-3.8B on What'sUp Subset B. For comparison, we also include the results for CLIP-ViT-L/14-336px and LLaVA-1.5-7B from the Table 1 in our submission.*
>     |                     | Left/Right, Indiv. | Left/Right, Pairs | Front/Behind, Indiv. | Front/Behind, Pairs |
>     | -------- |:-----------------------------:|:-------------------------------:|:-----------------------------:|:-------------------------------:|
>     | CLIP-ViT-L/14-336px |            54.9             |             10.8            |              51.5              |              7.8              |
>     | LLaVA-1.5-7B        |             **100**             |             **100**            |              **98.5**              |              **97.1**             |
>     | LLaMA-3-V-8B        |             71.1             |             46.1            |              69.1              |              41.2             |
>     | Phi-3-V-3.8B        |              **100**             |             **100**             |              56.9              |              13.7             |
>     | Random Chance       |          50.0          |          25.0          |          50.0         |          25.0         |
>
>     *Table 3: Results of LLaMA-3-V-8B and Phi-3-V-3.8B on What'sUp (four-way classification) benchmark. For comparison, we also include the results for CLIP-ViT-L/14-336px and LLaVA-1.5-7B from the Table 2 in our submission.*
>     |                   | Subset A, Indiv. | Subset A, Pairs | Subset A, Set of 4 | Subset B, Indiv. | Subset B, Pairs | Subset B, Set of 4 |
>     | -------- |:-----------------------------:|:-------------------------------:|:-----------------------------:|:-------------------------------:|:-----------------------------:|:-------------------------------:|
>     | CLIP-ViT-L/14-336px |           28.9       |       1.0       |         0.0        |       27.2       |       1.0       |         0.0        |
>     | LLaVA-1.5-7B        |            **62.1**       |       **41.3**      |        14.6        |       **74.0**       |       **61.8**      |        **23.5**        |
>     | LLaMA-3-V-8B        |           60.0       |       36.4      |        **17.5**        |       70.1       |       43.6      |        20.6        |
>     | Phi-3-V-3.8B        |             58.0       |       36.4      |        15.5        |       71.8       |       55.4      |        12.8        |
>     | Random Chance       |             25.0       |       6.3       |         0.4        |       25.0       |       6.3       |         0.4        |

---

> > ### Author Response · Authors · 2024-11-22
> > **Follow-up**
> >
> > 1. **How do these findings generalize to other MLLMs beyond LLaVA-1.5?** (Cont'd)
> >
> >     *Table 4: Results of LLaMA-3-V-8B and Phi-3-V-3.8B on COCO-spatial and GQA-spatial benchmark. For comparison, we also include the results for CLIP-ViT-L/14-336px and LLaVA-1.5-7B from the Table 2 in our submission.*
> >
> >     |            | COCO-spatial, one-obj. | COCO-spatial, two-obj. | GQA-spatial, one-obj. | GQA-spatial, two-obj. |
> >     | -------- |:-----------------------------:|:-------------------------------:|:-----------------------------:|:-------------------------------:|
> >     | CLIP-ViT-L/14-336px |          48.9          |          51.1          |          46.6         |          49.1         |
> >     | LLaVA-1.5-7B        |          96.0          |          82.3          |          96.0         |          90.7         |
> >     | LLaMA-3-V-8B        |          **97.8**          |          83.2          |          **99.0**         |          90.7         |
> >     | Phi-3-V-3.8B        |          97.3          |          **85.2**          |          98.0         |          **91.1**         |
> >     | Random Chance       |          50.0          |          50.0          |          50.0         |          50.0         |
> >
> >     Table 1~3 show that LLaMA-3-V-8B and Phi-3-V-3.8B can also extract distinct information from highly similar embeddings, though they are weak on some prepositions. These results show that our findings generalize to these two MLLMs with different scales and language models. We included these results in Appendix B.5 of our revised submission.
> >
> > 2. **What specific mechanisms allow MLLMs to extract distinct information from seemingly similar embeddings?**
> >
> >     In contrast to CLIP's dot product mechanism, MLLMs enable non-linear extraction of visual information from image embeddings and support complex interactions between image and text. For example, they assign different attention weights to image tokens and input text tokens when generating specific output text tokens. This flexibility allows the model to focus on distinct parts of the input, depending on the task at hand.

---

> > > ### Comment · Reviewer_2b8n · 2024-11-26
> > >
> > > I have read the author's response and the other rebuttal reviews. The work is very interesting, but I will maintain my score.

---

> > > > ### Author Response · Authors · 2024-11-27
> > > >
> > > > Thank you for taking the time to review our response and to read other reviews. We appreciate your feedback and are glad you found the work interesting. If there are any specific areas where we can provide further clarification or improvements, we’d be happy to address them. If the revision and our reply have addressed your concerns, we kindly ask you to reconsider your score.
> > > >
> > > > Thank you again for offering suggestions about our experiments and contributing to the discussion around our work.

---

### Official Review · Reviewer_fpLb · 2024-11-02

**Soundness:** 2
**Presentation:** 2
**Contribution:** 2
**Rating:** 3
**Confidence:** 5

**Summary:**

This paper provides a comprehensive study to analyze the answers supplied by the VLMs. Specifically, It compares the performances of CLIP and LlaVa-1.5-7B in the What’s Up and MMVP benchmarks. These benchmarks ask questions about a pair of images that contain the same objects and background but in different positions. This paper shows that the LlaVa-1.5-7B can perform better than CLIP in these benchmarks even when LlaVa uses CLIP as a visual encoder, and the average cosine similarity of the CLIP embedding of the image pair is greater than 0.95. Moreover, it provides ablation studies to explain this behavior.

**Strengths:**

This paper provides some interesting insights to show that the metric commonly used to measure the embedding similarity (Cosine Similarity) does not depict all aspects of vector pairs. Therefore, it suggested a complementary metric, Spearman’s rank correlation coefficient. However, table 1 only provides the average Cosine Similarity.

**Weaknesses:**

The paper is challenging to follow, primarily due to the absence of a clear statement of its main contributions in the Introduction. Its content closely parallels the CVPR24 paper, "Eyes Wide Shut? Exploring the Visual Shortcomings of Multimodal LLMs," raising concerns about the originality of this work. The CVPR24 paper highlights that Visual Language Models (VLMs), often relying on CLIP as the visual encoder, struggle with recognizing fine-grained details, such as object locations. It introduces the MMVP benchmark to evaluate these limitations comprehensively. I encourage the authors to clarify how their contributions provide novel insights beyond this existing research.

**Questions:**

I recommend including Spearman's rank correlation coefficient in Table 1 to enhance the analysis. Additionally, a more comprehensive study would be valuable. For example, could the authors provide Spearman's rank correlation coefficient and cosine similarity for the questions with the highest- and lowest-accurate answers?

---

> ### Author Response · Authors · 2024-11-21
> **Our Response to Your Concerns**
>
> Thank you for your comments and suggestions.
>
> First of all, we want to **point out a misunderstanding in your review**.
> In the "Strengths" section of your review, you concluded that,
> - *"Therefore, it suggested a complementary metric, Spearman's rank correlation coefficient."*
>
> However, Spearman's rank correlation coefficient is only used as a toy example in Section 3.2 for explaining why we hypothesize that distinct information might still be preserved in embeddings with high cosine similarity and can be extracted. This coefficient does not appear in other contexts in our paper, and is not part of our core contribution.
>
> After clarifying this, we address your concerns as follows:
> 1. **Your concerns on what our main contributions are and how our contributions provide novel insights beyond the CVPR 2024 paper.**
>
>     Our contributions are summarized as follows:
>
>     (1) First, we show evidence on the What'sUp benchmark that LLaVA-1.5 can distinguish images with CLIP embeddings of high cosine similarity, indicating that erroneous agreements are not the bottleneck of their visual reasoning performance on image pairs.
>
>     (2) Second, since we observe that CLIP has poor performance when image embeddings have high cosine similarity, we conduct ablation studies to see what difference between CLIP and LLaVA-1.5 causes their performance gap. We find that evaluation methods, language encoder, and training data are not the key factors that cause the gap, and we suggest that their difference in model paradigm (training and inference methods) plays an important role here.
>
>     (3) Third, we explore the true bottleneck of LLaVA-1.5 through two attempts in the discussion section. We find that LLaVA-1.5 does not attend enough to the visual input, and more visual information is preserved in CLIP image embeddings and aligned correctly than the pair accuracy suggested on the MMVP benchmark.
>
>     Our contribution (1) **counters the claim** in the CVPR2024 paper "Eyes Wide Shut" that MLLMs fail in simple questions on image pairs because their pre-trained CLIP vision encoders overlook crucial visual details in images as they encode them into highly similar embeddings. While this previous paper states that "vision models might become a bottleneck in multimodal systems," we suggest the potential of improving current MLLMs with fixed vision models through better utilizing visual information.
> 2. **Your suggestion on including Spearman's rank correlation in Table 1 and providing Spearman's rank correlation coefficient and cosine similarity for the questions with the highest- and lowest-accurate answers.**
>
>     As we explained at the begining, Spearman's rank correlation coefficient is not a metric we proposed to use on these benchmarks. It only serves as a toy example to show that cosine similarity does not depict all aspects of vector pairs, and we could still get different information from vectors with high cosine similarity. This motivates us to explore whether, in image embeddings with high cosine similarity (erroneous agreements), distinct information is actually preserved and can be extracted by MLLMs.
>
> We hope that our response could help you better understand the content and the contributions of our paper. If you have further questions or concerns, we are willing to address them as well!

---

> ### Comment · Reviewer_fpLb · 2024-11-26
>
> Thank you for the detailed response. After thoroughly reviewing your explanation and the feedback from other reviewers, I have identified additional concerns that warrant clarification.
>
> The performance comparison between LLaVA-1.5 and its vision encoder, OpenAI-CLIP-L/336, on the MMVP-VLM benchmark is interesting. However, the analysis appears limited, as it includes only a single comparison. It would be better if the authors provided a more diverse and consistent comparison, not limited to OpenAI-CLIP-L-336 vs. LLaVA-1.5, but also EVA-01-CLIP-g vs. InstructBLIP (Vicuna7b, Vicuna13b, Flan T5xl). In this way, the authors can analyze the influence of the different Visual Encoders and the LLM architectures on the performance boost.
>
> Furthermore, while it is interesting to observe improvements in the instructional model, the gains may result from LLaVA training. It adapts the language model (LLM) to leverage the frozen vision encoder better. This could enhance the LLM’s ability to extract visual information from sequence embeddings compared to CLIP models relying on the CLS token to align with smaller and limited language models. However, despite these improvements, the overall performance remains suboptimal. Consequently, I do not find the claim that “LLaVA-1.5 can distinguish images with CLIP embeddings of high cosine similarity, indicating that erroneous agreements are not the bottleneck of their visual reasoning performance on image pairs” to be fully substantiated.
>
> Given these considerations, I maintain my initial score.

---

> > ### Author Response · Authors · 2024-11-28
> > **Our Response to Your Further Concerns**
> >
> > Thank you for taking time to read our response and to read other reviews. We reply to your further concerns as follows:
> >
> > 1. **Your concern that our analysis is limited, as it only includes only a single comparison, and your suggestion on additional experiments.**
> >
> >     Thank you for your suggestion on providing a more diverse and consistent comparison using additional models. We would like to first clarify that we did not claim that all MLLMs outperform their visual components. For exploring whether the performance boosts generalize to other MLLM-image encoder pairs, in addition to LLaVA-1.5 (as well as LLaMA-3-V-8B and Phi-3-V-3.8B, which we included later in Appendix B.5), we evaluated two other MLLMs with distinct paradigms and training datasets: **InstructBLIP-Vicuna-7B** with EVA-CLIP-ViT-G/14 and **Otter-Image-MPT7B** with CLIP-ViT-L/14. These models were selected because they freeze the image encoder during training, aligning with the objectives of our comparison, unlike many MLLMs that finetune their image encoders. Meanwhile, they utilize the image embeddings in different ways from LLaVA-like models. The results on the What'sUp, COCO-spatial, and GQA-spatial benchmarks are as follows. We also included these results in Appendix B.5 of our revised submission.
> >
> >     *Table 1: Results of InstructBLIP and Otter on What'sUp Subset A. For comparison, we also include the results for CLIP-ViT-L/14-336px and LLaVA-1.5-7B from the Table 1 in our submission.*
> >
> >
> >     |    | Subset A, Left/Right, Indiv. | Subset A, Left/Right, Pairs | Subset A, On/Under, Indiv. | Subset A, On/Under, Pairs |
> >     | -------- |:---------------------:|:---------------------:|:-----------------:|:---------------------:|
> >     | CLIP-ViT-L/14-336px |           49.0             |             1.9             |            61.7            |            23.3           |
> >     | LLaVA-1.5-7B        |        **99.0**             |             **98.1**            |            80.1            |            60.2           |
> >     | InstructBLIP-Vicuna-7B        |   50.0             |             1.9            |            **93.7**            |            **87.4**           |
> >     | Otter-Image-MPT7B        | 50.0            |             1.0            |            56.8            |            13.6           |
> >     | Random Chance       |          50.0          |          25.0          |          50.0         |          25.0         |
> >
> >     *Table 2: Results of InstructBLIP and Otter on What'sUp Subset B. For comparison, we also include the results for CLIP-ViT-L/14-336px and LLaVA-1.5-7B from the Table 1 in our submission.*
> >
> >
> >    |   | Subset B, Left/Right, Indiv. | Subset B, Left/Right, Pairs | Subset B, Front/Behind, Indiv. | Subset B, Front/Behind, Pairs |
> >     | -------- |:------------:|:---------:|:------------:|:------------:|
> >     | CLIP-ViT-L/14-336px |            54.9             |             10.8            |              51.5              |              7.8              |
> >     | LLaVA-1.5-7B        |             **100**             |             **100**            |              **98.5**              |              **97.1**             |
> >     | InstructBLIP-Vicuna-7B       |             50.0             |             0.0            |              50.0              |              5.9             |
> >     | Otter-Image-MPT7B        |              50.0             |             0.0             |              51.5              |              11.8             |
> >     | Random Chance       |          50.0          |          25.0          |          50.0         |          25.0         |
> >
> >     *Table 3: Results of InstructBLIP and Otter on What'sUp (four-way classification) benchmark. For comparison, we also include the results for CLIP-ViT-L/14-336px and LLaVA-1.5-7B from the Table 2 in our submission.*
> >
> >     |           | Subset A, Indiv. | Subset A, Pairs | Subset A, Set of 4 | Subset B, Indiv. | Subset B, Pairs | Subset B, Set of 4 |
> >     | -------- |:-------:|:------:|:--------:|:----------:|:--------:|:---------:|
> >     | CLIP-ViT-L/14-336px |           28.9       |       1.0       |         0.0        |       27.2       |       1.0       |         0.0        |
> >     | LLaVA-1.5-7B        |            **62.1**       |       **41.3**      |        **14.6**        |       **74.0**       |       **61.8**      |        **23.5**        |
> >     |  InstructBLIP-Vicuna-7B        |           37.6       |       25.7      |        0.0        |       29.9       |       15.2      |        0.0        |
> >     | Otter-Image-MPT7B        |             24.5       |       2.4      |        0.0        |       24.8       |       3.0      |        0.0        |
> >     | Random Chance       |             25.0       |       6.3       |         0.4        |       25.0       |       6.3       |         0.4        |

---

> > > ### Author Response · Authors · 2024-11-28
> > > **Follow-up**
> > >
> > > 1. **Your concern that our analysis is limited, as it only includes only a single comparison, and your suggestion on additional experiments.** (Cont'd)
> > >
> > >    *Table 4: Results of InstructBLIP and Otter on COCO-spatial and GQA-spatial benchmark. For comparison, we also include the results for CLIP-ViT-L/14-336px and LLaVA-1.5-7B from the Table 2 in our submission.*
> > >
> > >    |                     | COCO-spatial, one-obj. | COCO-spatial, two-obj. | GQA-spatial, one-obj. | GQA-spatial, two-obj. |
> > >     | -------- |:-----------------------------:|:-------------------------------:|:-----------------------------:|:-------------------------------:|
> > >     | CLIP-ViT-L/14-336px |          48.9          |          51.1          |          46.6         |          49.1         |
> > >     | LLaVA-1.5-7B        |          **96.0**          |          **82.3**          |          **96.0**         |         **90.7**         |
> > >     |  InstructBLIP-Vicuna-7B        |          55.0          |          51.4          |         47.8         |          50.2         |
> > >     | Otter-Image-MPT7B        |          51.9          |          50.0          |          54.1         |          51.9         |
> > >     | Random Chance       |          50.0          |          50.0          |          50.0         |          50.0         |
> > >
> > >    Compared with LLaVA-1.5, with different architectures and training data, Otter and InstructBLIP struggle on this benchmark (close to random chance except On/Under for InstructBLIP). Due to their poor performance, we did not include the evaluation of their visual components here. Hence, we can see that MLLMs do not guarantee more effective extraction from frozen image encoder. Good design of MLLM architecture and training data synergize to provide strong visual information extraction ability.
> > >
> > > 2. **Your concern that our claim on LLaVA-1.5 is not fully substantiated.**
> > >
> > >     Our finding that LLaVA-1.5 performs well on image pairs with CLIP embeddings of high cosine similarity emphasizes the **feasibility** of extracting distinct information from such pairs using MLLMs, but not any guarantee of MLLMs to perform well on all similar image pairs. There are many factors beyond image encoder at play in MLLM's visual reasoning, such as language model's reasoning ability.
> > >
> > >     This feasibility reveals that highly similar CLIP embeddings are not the main culprit of the suboptimality of LLaVA-1.5 on MMVP benchmark, and there should be other causes. As an initial exploration of these causes, our experiments in discussion section show that visual information is not fully utilized in answer generation, implying the possibility of improving model performance with image encoder fixed.
> > >
> > > If you have further questions or concerns, we would be happy to address them. Thank you again for your feedback!

---

> > > > ### Comment · Reviewer_fpLb · 2024-11-28
> > > >
> > > > I thank the authors for their great effort in providing these new results. However, I consider it a failure in the evaluation protocol. The novel experiments are not comparable with OpenAI-CLIP-ViT-L/14-336 because, as the author outlined, they leverage different visual encoders. InstructBLIP-Vicuna-7B uses EVA-CLIP-ViT-G/14, Otter-Image-MPT7B uses CLIP-ViT-L/14. If the authors want to analyze these methods, they should also provide the results of EVA-CLIP-ViT-G/14 and CLIP-ViT-L/14. Moreover, it is worth noting that EVA-CLIP-ViT-G/14 is better than OpenAI-CLIP-ViT-L-336, as seen in the CVPR2024 paper "Eyes Wide Shut". So, EVA-CLIP could surpass the OpenAI-CLIP-ViT-L-336 and even match the InstructBLIP, indicating that the phenomena the authors highlight could be particular to LLaVA.

---

> > > > > ### Author Response · Authors · 2024-11-29
> > > > > **Response to Your Further Concerns**
> > > > >
> > > > > Thank you for checking our new response!
> > > > >
> > > > > First of all, as we clarified in our previous replies, we would like to reiterate that we did not claim that MLLMs are guaranteed to outperform their visual components. Instead, our argument in the paper is that LLaVA-1.5 is able to extract distinct information from highly similar CLIP embeddings, so erroneous agreements are not the main culprit of its failure on image pairs with such property (e.g., the MMVP benchmark).
> > > > >
> > > > > As for your concerns, we responded as follows.
> > > > >
> > > > > - *"However, I consider it a failure in the evaluation protocol ... If the authors want to analyze these methods, they should also provide the results of EVA-CLIP-ViT-G/14 and CLIP-ViT-L/14."*
> > > > >
> > > > >     The reason why we did not test the results for EVA-CLIP-ViT-G/14 in our previous reply is that we already observed low performance of InstructBLIP and Otter on the What'sUp benchmark. (Their performances on most tasks we evaluated on are below or close to random chance as you can see from the tables.) So they could not outperform their image encoder on these benchmarks.
> > > > >
> > > > >     If you are interested, we now provide the results for CLIP-ViT-L/14 and EVA-CLIP-ViT-G/14 below. The results for CLIP-ViT-L/14 correspond to Table 2 of our submission, while those for EVA-CLIP-ViT-G/14 were obtained using the model pretrained on a filtered version of LAION-400M, as provided by the OpenCLIP repository. We can see that these image encoders have performance below or close to random chance on these benchmarks.
> > > > >
> > > > >     *Table 1: Results of CLIP-ViT-L/14 and EVA-CLIP-ViT-G/14 on What'sUp Subset A. For comparison, we also include the results for CLIP-ViT-L/14-336px.*
> > > > >     |  | Subset A, Left/Right, Indiv. | Subset A, Left/Right, Pairs | Subset A, On/Under, Indiv. | Subset A, On/Under, Pairs |
> > > > >     | ----- |:-----:|:---------:|:--------:|:----------:|
> > > > >     | CLIP-ViT-L/14-336px |  49.0   |   1.9  |   61.7 | 23.3    |
> > > > >     | CLIP-ViT-L/14 |  49.0  | 2.9  |  60.2 | 21.4 |
> > > > >     | EVA-CLIP-ViT-G/14  |  49.0  | 1.0| 56.3 | 14.6  |
> > > > >     | Random Chance |  50.0  | 25.0 | 50.0  |   25.0 |
> > > > >
> > > > >     *Table 2: Results of CLIP-ViT-L/14 and EVA-CLIP-ViT-G/14 on What'sUp Subset B. For comparison, we also include the results for CLIP-ViT-L/14-336px.*
> > > > >     |   | Subset B, Left/Right, Indiv. | Subset B, Left/Right, Pairs | Subset B, Front/Behind, Indiv. | Subset B, Front/Behind, Pairs |
> > > > >     | ---- |:------:|:---------:|:-------:|:---------:|
> > > > >     | CLIP-ViT-L/14-336px |    54.9     |  10.8 | 51.5 | 7.8    |
> > > > >     | CLIP-ViT-L/14 |   54.9     |    11.8| 51.0      |      9.8        |
> > > > >     | EVA-CLIP-ViT-G/14  |  50.1  | 4.9 | 52.9     |     14.7        |
> > > > >     | Random Chance    |    50.0     |   25.0   |  50.0 |   25.0  |
> > > > >
> > > > >     *Table 3: Results of CLIP-ViT-L/14 and EVA-CLIP-ViT-G/14 on What'sUp (four-way classification) benchmark. For comparison, we also include the results for CLIP-ViT-L/14-336px.*
> > > > >     |     | Subset A, Indiv. | Subset A, Pairs | Subset A, Set of 4 | Subset B, Indiv. | Subset B, Pairs | Subset B, Set of 4 |
> > > > >     | -------- |:---------:|:------:|:-----:|:---------:|:----------:|:----------:|
> > > > >     | CLIP-ViT-L/14-336px | 28.9  |   1.0  |         0.0        |       27.2       |       1.0       |         0.0        |
> > > > >     | CLIP-ViT-L/14 |  26.7  |   1.0       |         0.0        |       25.7       |       1.5       |         0.0        |
> > > > >     | EVA-CLIP-ViT-G/14   |     28.2     |     2.4     |        0.0      |     27.9      |    5.4   |    0.0     |
> > > > >     | Random Chance       |   25.0 |    6.3       |         0.4        |       25.0       |       6.3       |         0.4        |
> > > > >
> > > > >     *Table 4: Results of CLIP-ViT-L/14 and EVA-CLIP-ViT-G/14 on COCO-spatial and GQA-spatial benchmark. For comparison, we also include the results for CLIP-ViT-L/14-336px.*
> > > > >
> > > > >     |    | COCO-spatial, one-obj. | COCO-spatial, two-obj. | GQA-spatial, one-obj. | GQA-spatial, two-obj. |
> > > > >     | -------- |:---------:|:------:|:----------:|:--------------:|
> > > > >     | CLIP-ViT-L/14-336px |          48.9          |          51.1          |          46.6         |          49.1         |
> > > > >     | CLIP-ViT-L/14 |         49.1         |       50.2          |          46.0        |          48.1         |
> > > > >     | EVA-CLIP-ViT-G/14        |      45.9        |          50.5         |      44.4       |      49.8       |
> > > > >     | Random Chance       |          50.0          |          50.0          |          50.0         |          50.0         |
> > > > >
> > > > > - *"EVA-CLIP could surpass the OpenAI-CLIP-ViT-L-336 and even match the InstructBLIP, indicating that the phenomena the authors highlight could be particular to LLaVA."*
> > > > >
> > > > >     (1) From the results above, we did not observe that EVA-CLIP surpasses InstructBLIP on these benchmarks.
> > > > >
> > > > >     (2) Regardless of whether EVA-CLIP surpasses InstructBLIP, our conclusion remains unchanged, as we do not claim that all MLLMs outperform their visual components.
> > > > >
> > > > > If you have further questions or concerns, we are willing to address them!

---

### Author Response · Authors · 2024-11-22
**Summary of the Reviews, Responses, and Changes to the Paper**

We thank the reviewers for their comments and suggestions regarding our submission. We appreciated positive comments on our contributions, e.g., the paper "challenges and refines an important assumption in the field about VLM limitations" from Reviewer `2b8n`, and it "delivers a valuable message to the community ... counters that (previous) claim" from Reviewer `DMVn`. We also found that reviewers gave positive feedbacks on our ablation studies with analysis (Reviewer `7vBu`) and the interesting observation in the discussion (Reviewer `7vBu` and Reviewer `DMVn`).

We also thank the reviewers for their constructive criticisms. Below, we summarize the major issues and our responses to them:

### **Concerns on the storyline being unclear and confusing from Reviewer `DMVn` and Reviewer `fpLb`.**

Our storyline is as follows: Previous research attributes LLaVA-1.5's low MMVP performance to deficiencies in the CLIP vision encoder (namely erroneous agreements). We show that LLaVA-1.5 (which uses the CLIP vision encoder) excels on image pairs with erroneous agreements on What'sUp benchmark, whereas CLIP model (using both vision encoder and text encoder) has a poor performance. Through ablation, we pinpoint the key differences between LLaVA-1.5 and CLIP that drives this gap and revisit MMVP benchmark to uncover the true cause of LLaVA-1.5's shortcomings.

From the feedback of Reviewer `DMVn`, we found that the What'sUp and MMVP benchmark and model performance on them might be unfamiliar to the readers and potentially causes confusion. So we added more description in the introduction to indicate that LLaVA-1.5 perform well on What'sUp but not on more challenging MMVP, guiding the readers to better understand the storyline.

### **Confusion about the function and description of the Spearman's rank correlation from Reviewer `7vBu` and Reviewer `fpLb`.**

*First, Spearman's rank correlation only serves as a toy example and does not appear in any other contexts in our submission*. It motivates our exploration and comparison using LLaVA-1.5 on extracting information from highly similar embeddings. However, we did not propose to put it into use in experiments and it is not part of our core contribution.

Second, for Reviewer `7vBu`'s question about how high negative Spearman's rank correlation show that the embeddings are "fully opposed," we did not intend to show that the vectors are fully opposed in the sense of cosine similarity. Instead, they contain opposed order information. We clarify this by replacing "they are opposed in this sense" with "their order information is fully opposed" in the new version, which better conveys our intended meaning.

## Summary of changes to the draft

We have made several changes to the draft based on the reviewers' suggestions (all changes are highlighted in blue in the updated PDF). Specifically:

- In response to Reviewer `2b8n`'s concern about the generalizability of our findings beyond LLaVA-1.5, we included evaluations on What'sUp for two other MLLMs (LLaMA-3-V-8B and Phi-3-V-3.8B) in Appendix B.5, demonstrating that our findings in Section 3 generalize to these models as well.
- For concerns on the storyline raised by Reviewer `DMVn`, we refined the Figure 1 and its caption, and added descriptions in introduction (L52-L54, L67) and Section 3 (L193-L195).
- For Reviewer `DMVn`'s concerns about the scope and dataset choice, we added more captions to Figure 2 and included our reasoning in the related work section (L104–L107).
- To clarify the meaning of "visual information extraction" and the toy example as suggested by Reviewer `7vBu`, we rephrased descriptions in the introduction (L74–L76) and Section 3 (L243–L244).
- We combined the original Table 5 and Table 6 to save space and improve readability.

---

> ### Author Response · Authors · 2024-11-28
> **Follow-up**
>
> In response to Reviewer `fpLb`'s suggestion on providing a more diverse and consistent comparison (not limited to OpenAI-CLIP-L-336 vs. LLaVA-1.5), we further updated the Appendix B.5 with evaluation for InstructBLIP-Vicuna-7B and Otter-Image-MPT7B on What'sUp, COCO-spatial, and GQA-spatial benchmark.

---

### Meta-Review · Area_Chair_kLgE · 2024-12-21

**Metareview:**

This paper investigates whether CLIP's image embeddings are truly the bottleneck for vision-language models' (VLMs) performance on visual reasoning tasks. Using LLaVA-1.5 as the primary example, the authors challenge the previous assumption that erroneous agreements in CLIP embeddings (high cosine similarity between visually distinct images) are responsible for VLMs' poor visual reasoning performance.

### Strengths:

1. Novel perspective challenging existing assumptions:
> "delivers a valuable message to the community ... counters that claim, attributing the problem instead to the model not utilizing these visual features effectively" - Reviewer DMVn

2. Systematic experimental methodology:
> "Demonstrates that existing architectures might be more capable than previously thought, just requiring better utilization strategies" - Reviewer 2b8n

3. Technically sound ablation studies:
> "Important analysis shown in section 4 (Investigating the performance gap)... visual information extraction is the key factor in determining the performance gap" - Reviewer 7vBu

## Weaknesses:

1. Limited scope and generalizability:
> "WhatsUp, MMVP, COCO-spatial and GQA-spatial are not really well-known datasets... These datasets are not enough to reflect model reasoning and to come up with general conclusions" - Reviewer DMVn

2. Methodological concerns:
> "The novel experiments are not comparable with OpenAI-CLIP-ViT-L/14-336 because, as the author outlined, they leverage different visual encoders" - Reviewer fpLb

3. Insufficient evidence for claims:
> "I find the claim that the authors make not well-supported by experimental evidence. The benchmarks the authors used are not enough to support this claim" - Reviewer DMVn

### Justification

Despite interesting insights and thorough revisions, fundamental concerns about methodology, generalizability, and evidence remain. All reviewers rated the paper below the acceptance threshold (three at "marginally below" and one "reject"), indicating the contribution does not meet the conference bar. Key issues include:

1. Limited benchmark selection that doesn't fully support the claims
2. Methodological issues in model comparisons
3. Lack of evidence that findings generalize beyond LLaVA


While the paper raises important questions about VLM limitations and CLIP embeddings, the methodological concerns and limited scope make it unsuitable for acceptance at this time. The authors' revisions, while thorough, did not fully address these core issues as evidenced by maintained negative ratings from all reviewers.

**Additional Comments On Reviewer Discussion:**

The authors made significant efforts to address reviewer concerns:

1. Added experiments with additional models (LLaMA-3-V-8B, Phi-3-V-3.8B, InstructBLIP, Otter-Image)
2. Included NaturalBench evaluation for generalizability
3. Clarified methodology and scope

However, none of the reviewers changed their scores after the revisions:
- fpLb maintained that experiments remain incomparable
- DMVn still found evidence insufficient
- 2b8n and 7vBu maintained their "marginally below threshold" ratings

---

### Decision · Program_Chairs · 2025-01-22

Reject